# CRAFT: Time Series Forecasting with Cross-Future Behavior Awareness

## Abstract

Time series forecasting is the crucial infrastructure in the field of e-commerce, providing technical support for consumer behavior analysis, sales trends forecasting, etc. E-commerce allows consumers to reserve in advance. These pre-booking features reflect future sales trends and can increase the certainty of time series forecasting issues. In this paper, we define these features as Cross-Future Behavior, which occurs before the current time but takes effect in the future. To increase the performance of time series forecasting, we leverage these features and propose the **CR**oss-Future Behavior **A**wareness based **T**ime Series **F**orecasting method (CRAFT). The core idea of CRAFT is to utilize the trend of cross-future behavior to mine the trend of time series data to be predicted. Specifically, to settle the sparse and partial flaws of cross-future behavior, CRAFT employs the Koopman Predictor Module to extract the key trend and the Internal Trend Mining Module to supplement the unknown area of the cross-future behavior matrix. Then, we introduce the External Trend Guide Module with a hierarchical structure to acquire more representative trends from higher levels. Finally, we apply the demand-constrained loss to calibrate the distribution deviation of prediction results. We conduct experiments on real-world dataset. Experiments on both offline large-scale dataset and online A/B test demonstrate the effectiveness of CRAFT. Our dataset and code will be released after formal publication.

## 1 Introduction

Time series forecasting (TSF) is the crucial infrastructure in e-commerce (Ryali et al., 2023). By analyzing trends and patterns in e-commerce data, TSF allows businesses to gain insights into both operational behaviors and customer dynamics over time, thereby supporting consumer behavior analysis and sales trend predictions. However, accurate TSF is a challenging task given the need to model complex, non-linear temporal patterns over long periods of time (Rasul et al., 2024). To this end, researchers explore various backbone networks such as convolutional neural networks (CNNs) (Chen et al., 2020), recurrent neural networks (RNNs) (Yin et al., 2022), and Transformer (Zhou et al., 2021). Additionally, they work on incorporating richer features into TSF and study issues related to multivariate TSF (Zhao & Shen, 2024; Li et al., 2024) and feature decomposition (Zeng et al., 2023; Liu et al., 2024).

With the advancement of the Internet, e-commerce has introduced the pre-sale retail model. In many scenarios, such as hotel check-ins and purchasing airline or tourist attraction tickets, customers can book products in advance and consume the items they have booked later. This process can produce useful priori information for TSF. In this paper, we defined the in-advanced booking information as **Cross-Future Behavior (CFB)**: features that occur before the current time but take effect in the future. CFB positively affects TSF. As shown in Figure 1, if only relying on the trend of the label, the future trend may be predicted as a downward trend. With the guidance of CFB, the prediction result will be more precise.

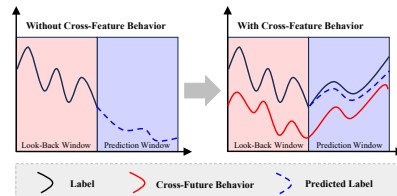

Figure 1: Illustration of Cross-Future Behavior (CFB).

In fact, existing work is also attempting to introduce more features into TSF. Zeng et al. (2023) attempts to extract more information from the acquired data and decomposes the time series into trend and remainder parts. Wen et al. (2017) classifies covariate features in TSF into dynamic historical, known future, and static variables. CFB, we proposed in this paper, can be treated as the known future variable and has many excellent properties. CFB contains true information related to future events. The future trend of the prediction target, even the abnormal trend caused by sudden events, can be reflected in the trend of CFB. However, as the existing TSF models primarily focus on exploring the correlation between the historical series and future trends (Chen et al., 2020; Yin et al., 2022; Zhou et al., 2021), it is difficult to achieve TSF that integrates CFB through existing models. CFB-based TSF faces two main challenges. **1) CFB is sparse and partial.** Consumers can book items at any time, causing CFB to remain fully observed until the last minute. Thus, if CFB

is simply incorporated into the model, the prediction model may be unable to apply CFB features correctly and even make incorrect predictions due to CFB. **2) The trend of CFB is unobvious.** As shown in Figure 2, compared with the sales trend in a business district, the sales trend in an individual hotel is unobvious. CFB has the same nature, and the trend in an individual hotel is much unobvious compared with that in a high-level business district. Consequently, devising a method to utilize the sales trend in high level to inform the auxiliary forecast for an individual hotel presents another significant challenge.

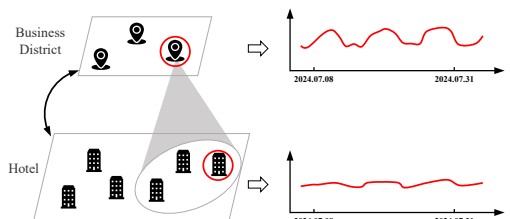

Figure 2: Sales trend comparison between the business district and an individual hotel.

Therefore, jointly considering the above challenges, we propose CRAFT, a **Cr**oss-Future Behavior **A**wareness based **T**ime Series **F**orecasting method, for the first time to utilize Cross-Future Behavior to realize time series forecasting. The core idea of CRAFT, as shown in Figure 1, is to utilize the trend of CFB to mine the trend of time series data to be predicted. CRAFT is composed of three main parts: the Koopman Predictor Module (KPM), the Internal Trend Mining Module (ITM), and the External Trend Guide Module (ETG). KPM can extract the key trends of the label and CFB, predicting the label in the prediction window. ITM supplements the unknown area of CFB, making the final prediction of the label in the prediction window. ETG, with a hierarchical structure, can acquire more representative trends from higher levels. Finally, we apply the demand-constrained loss to calibrate the distribution deviation of prediction results. We conduct experiments on real-world dataset. Experiments on both offline large-scale dataset and online A/B test demonstrate the effectiveness of CRAFT. We summarize the main contributions of this paper as follows:

- We define the Cross-Future Behavior (CFB) and apply the CFB feature to time series forecasting for the first time. CFB is a feature discovered from our extensive real case studies and has superior characteristics: the trend of CFB can reflect the prediction target and even the abnormal trend of the target.
- We propose a novel framework, namely CRAFT, to realize CFB-based time series forecasting. CRAFT can utilize the trend of Cross-Future Behavior to mine the trend of prediction targets. CRAFT is composed of three main modules, including KPM, ITM, and ETG, to address the two challenges when applying CFB to time series forecasting. KPM and ITM can address the sparse and partial flaws of CFB, and ETG can address the unobvious trend flaws of CFB.
- Extensive offline experiments on the real-world dataset and online A/B tests show the superiority of CRAFT towards SOTA baselines. Specifically, CRAFT improves application performance significantly, with an improvement rate of $41.35\%$ on the $IWR$ metric. Currently, CRAFT has been successfully deployed on the reality application, serving online hotel inventory negotiations.

## 2 RELATED WORK

**Backbone for time series forecasting.** Recently, Transformer has reshaped the landscape of machine learning across numerous fields. Many works attempt to apply Transformer to forecast time series data (Wen et al., 2023). PatchTST (Nie et al., 2024) designs a channel-independent Transformer for time series forecasting. To address the channel-independent limitations of Transformer, CARD (Wang et al., 2024) proposes a channel-aligned attention structure that can acquire both tem-

poral correlations and dynamical dependence among multiple variables over time. Informer (Zhou et al., 2021) extends Transformer using ProbSparse based on KL divergence to solve Long Sequence time series forecasting. TFT (Lim et al., 2021) introduces a novel attention-based architecture that combines high-performance multi-horizon forecasting with interpretable insights into temporal dynamics. Autoformer (Wu et al., 2021) proposes a Decomposition Architecture and Auto-Correlation Mechanism based on stochastic process theory to realize the series-wise connection and break the bottleneck of information utilization. Pyraformer (Liu et al., 2021) proposes a new Transformer based on a pyramidal attention module to simultaneously capture temporal dependencies of different ranges in a compact multi-resolution fashion. Besides Transformer, a variety of other network architectures are widely investigated. Convolutional neural networks-based time series forecasting models such as WaveNet (Oord et al., 2016), TCN (Bai et al., 1803), and DeepTCN (Chen et al., 2020) use causal convolution to learn sequences and use dilated convolution and residual block to memorize historical patterns. Graph WaveNet (Zonghan et al., 2019) enhances the WaveNet framework by using an adaptive and learnable adjacency matrix to automatically infer graph structures, enabling the prediction of spatiotemporal sequences. Moreover, due to the sequential nature of time series data, Recurrent Neural Networks-based time series forecasting is particularly widely suited, mainly modeling the temporal dependence of time series (Salinas et al., 2020; Rangapuram et al., 2018; Wen et al., 2017; Wang et al., 2019; Liu et al., 2020).

**Multivariate time series forecasting.** Multivariate time series forecasting utilizes multiple time-dependent variables to realize prediction. Compared with univariate time series forecasting, multivariate time series forecasting can help to better understand the interactions between different components of a complex system, which is crucial for strategy formulation and decision-making (Mendis et al., 2024). There are two commonly used strategies in multivariate TSF, i.e., the channel-dependent (CD) and channel-independent (CI) methods. CI method only models cross-time dependence, and the CD method models both cross-time dependence and cross-variate dependence (Zhao & Shen, 2024). While the CI method is characterized by simplicity and low risk of overfitting, the CD method has inevitably become the mainstream of research. Recently, Zhao & Shen (2024) utilizes the channel dependence between variates and proposes a plug-and-play method named LIFT, which exploits the lead-lag relationship between variates by estimating leading indicators and leading steps. LIFT refreshes the accuracy of multivariate TSF.

**Feature decomposition in time series forecasting.** Different from the multivariate TSF that utilizes multiple variates and the dependence between variates to realize prediction, feature decomposition in TSF does not introduce new variates. The core idea of feature decomposition is to extract as much information as possible from existing variates. Dlinear (Zeng et al., 2023) decomposes time series into trend series and remainder series and uses two single-layer linear networks to model them, bringing performance improvements. Koopa (Liu et al., 2024) solves non-stationary time series prediction problems from the perspective of modern dynamics Koopman theory.

## 3 PRELIMINARIES

Time series forecasting (TSF) with Cross-Future Behavior (CFB) can be defined as $\mathbf{Y}_{t+1:t+P} = H(\mathbf{Y}_{t-L+1:t}, \mathbf{X}_{\mathbb{T}}, \mathbf{C}_{\mathbb{T}})$. $\mathbf{Y}_{t-L+1:t} \in \mathbb{R}^L$ and $\mathbf{Y}_{t+1:t+P} \in \mathbb{R}^P$ are time series data (i.e., label) at the $L$-length look-back window and $P$-length prediction window at time $t$ respectively. $\mathbf{X}_{\mathbb{T}}$ is covariate features, $\mathbf{C}_{\mathbb{T}}$ is the CFB feature and $H$ is the prediction function to be learned. The covariate features (Wen et al., 2017) $\mathbf{X}_t$ contains three categories: 1) historical features like month-on-month sales features, etc; 2) known future features like holidays, weekends, etc; 3) static features like hotel brands, business districts, etc. $\mathbf{C}_{\mathbb{T}} = \{\mathbf{C}_{t-L+1:t}, \mathbf{C}_{t+1:t+P}\}$, where $\mathbf{C}_{t-L+1:t}$ is CFB in the look-back window and $\mathbf{C}_{t+1:t+P}$ is CFB in the prediction window. It is worth noting that $\mathbf{C}_{t+1:t+P}$ in the prediction window is partial as this is a not fully observable variate. Consumers can book items in the prediction window at any time until the last minute. More detailed introduction to CFB feature $\mathbf{C}_t$ refers to Appendix A.

In the following section, we omit the subscripts of some symbols for simplicity. Specifically, we denote time series at the look-back window $\mathbf{Y}_{t-L+1:t}$ as $\mathbf{Y}_L$, time series at the prediction window $\mathbf{Y}_{t+1:t+P}$ as $\mathbf{Y}_P$, CFB feature in look-back window $\mathbf{C}_{t-L+1:t}$ as $\mathbf{C}_L$, CFB feature in prediction window $\mathbf{C}_{t+1:t+P}$ as $\mathbf{C}_P$. In addition, as $\mathbf{C}_P$ is partial observed, we define a new notion $\mathbf{C}_{TP}$ to indicate the ground truth of CFB in prediction window. For clarity, we summarize all notions with table in Appendix B.1.

# 4 METHODOLOGY

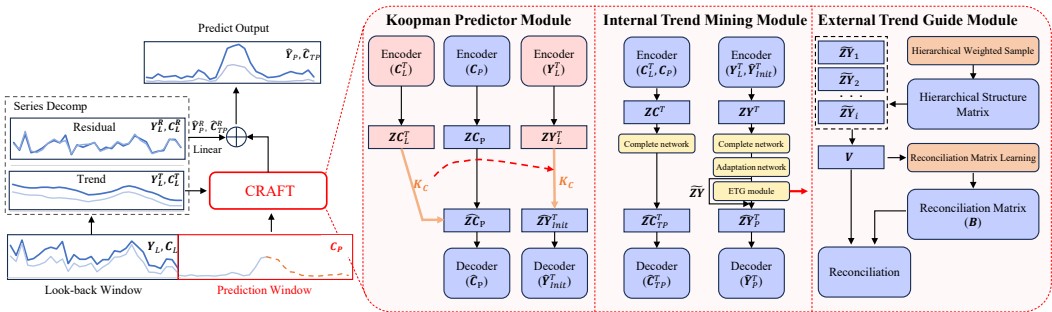

Figure 3: The overview of the proposed **Cr**oss-Future Behavior **A**wareness based **T**ime Series **F**orecasting method (CRAFT). The left decomposition part is based on DLinear (Zeng et al., 2023). CRAFT is composed of three main parts, the Koopman Predictor Module (KPM), the Internal Trend Mining Module (ITM), and the External Trend Guide Module (ETG). KPM is used to extract the key trends of the trend of label and CFB, predicting the label in the prediction window. ITM is used to supplement the unknown area of the CFB. ETG is used to acquire more representative trends from higher levels.

Figure 3 depicts the overview of the proposed CRAFT method. CRAFT uses DLinear (Zeng et al., 2023) to decompose the time series data $\mathbf{Y}_L, \mathbf{C}_L$ in the look-back window into the trend $\mathbf{Y}_L^T, \mathbf{C}_L^T$ and residual $\mathbf{Y}_L^R = \mathbf{Y}_L - \mathbf{Y}_L^T, \mathbf{C}_L^R = \mathbf{C}_L - \mathbf{C}_L^T$ components. The moving average kernel with a certain kernel size is used in the decomposition process. CRAFT's core idea lies in how to use the trend of partial CFB to mine the trend of label. We prove the consistency between partial CFB, CFB, and label theoretically in Appendix D.1 and D.2. CRAFT is composed of three submodules: Koopman Predictor Module (KPM), Internal Trend Mining Module (ITM), and External Trend Guide Module (ETG). **KPM** employs the Koopman operator to linearly map the CFB features from the look-back window to the prediction window, after projecting the raw data into the mapping space. Subsequently, it supervises the initialization of the label sequence within the prediction window. **ITM** completes linear mapping between the look-back window and prediction window and adopts an adaptation operator to instruct the evolution of the time series of the label sequentially. Due to higher-level temporal sequences having lower noise and stronger regularity compared to lower-level temporal sequences, **ETG** module makes forecast results more robust by structuring the reconciliation matrix and calibrating the predicted outcomes of root nodes. To avoid computational problems caused by excessively large hierarchical matrices, the concept of hierarchical sampling is introduced.

## 4.1 KOOPMAN PREDICTOR MODULE

**KPM** module aims to transfer the future trend information from CFB feature $\mathbf{C}_{\mathbb{T}} = \{\mathbf{C}_L, \mathbf{C}_P\}$ to labels $\mathbf{Y}_P$. The simplified framework of KPM is shown in Figure 3 and the specific framework refers to Appendix C.1. KPM employs an encoder-decoder framework with the input of CFB $\mathbf{C}_L$ and label $\mathbf{Y}_L$ in the look-back window and output of partial CFB $\mathbf{C}_P$ in the prediction window. Concretely, we first construct an encoder $\mathbb{R}^L \mapsto \mathbb{R}^D$ as a data-driven measurement function, i.e., $g(x)$ in Appendix B.2. The encoder module is Multi Layer Perception (MLP) (Zhu et al., 2023) and it can also be replaced to other structure: $\mathbf{ZC}_L^T = Encoder(\mathbf{C}_L^T), \mathbf{ZC}_P = Encoder(\mathbf{C}_P), \mathbf{ZY}_L^T = Encoder(\mathbf{Y}_L^T)$, where $\mathbf{ZC}_L^T, \mathbf{ZC}_P, \mathbf{ZY}_L^T$ are the embeddings of the trend of CFB in the look-back window, CFB in the prediction window, and the trend of labels in the look-back window. In particular, the encoder is shared for $\mathbf{ZC}_L^T, \mathbf{ZC}_P, \mathbf{ZY}_L^T$. Secondly, based on the Koopman Theory (Koopman, 1931), we use finite linear matrix $\mathbf{K}_C$ to approach infinite koopman matrix $\mathcal{K}$ to simulate the evolution process between time periods, that is:

$$\hat{\mathbf{ZC}}_P = \mathbf{K}_C \times \mathbf{ZC}_L^T, \tag{1}$$

where $\mathbf{K}_C \in \mathbb{R}^{D \times D}$ is the Koopman matrix for CFB, which contains the future sales trend information. The feasibility proof of $\mathbf{K}_C$ is in Appendix D.1. Referring to Eq. (17), its value can be calculated as:

$$\mathbf{K}_C = (\mathbf{ZC}_L^{T\top} \times \mathbf{ZC}_L^T + \lambda \mathbf{E})^{-1} \times \mathbf{ZC}_L^{T\top} \times \hat{\mathbf{ZC}}_P. \tag{2}$$

Then, we use $\mathbf{K}_C$ to convert the future trend information from CFB embedding to label embedding:

$$\mathbf{Z}\hat{\mathbf{Y}}_{Init}^T = \mathbf{K}_C \times \mathbf{Z}\mathbf{Y}_L^T. \tag{3}$$

Finally, we construct a decoder $\mathbb{R}^D \mapsto \mathbb{R}^P$ to obtain the preliminary prediction result of label. Same as encoder, decoder can adopt various model structures, and this paper uses the MLP layer (Zhu et al., 2023). In particular, the decoder is shared for $\hat{\mathbf{C}}_P$, $\hat{\mathbf{Y}}_{Init}^T$: $\hat{\mathbf{C}}_P = Decoder(\mathbf{Z}\hat{\mathbf{C}}_P)$, $\hat{\mathbf{Y}}_{Init}^T = Decoder(\mathbf{Z}\hat{\mathbf{Y}}_{Init}^T)$. Specifically, to ensure that $\mathbf{K}_C$ is meaningful, we construct a recovery loss $\mathcal{L}_{be\_k}$ to constrain the decoder to restore the original data based on the embedding output by the encoder, so that the embedded latent variable enables to obtain the potential attributes from raw data and preserve the original information as much as possible. $\mathcal{L}_{be\_k}$ is designed based on the MSE loss:

$$\mathcal{L}_{be\_k} = \frac{2}{P^2} \sum_{i=0}^{P} \sum_{j=0}^{i} (\hat{\mathbf{C}}_P[i,j] - \mathbf{C}_P[i,j])^2, \tag{4}$$

we choose $\mathbf{C}_P$ to calculate loss because we emphasize more on tendency characterization at the prediction window. It should be noted that we only focus on the known parts, i.e., $j \leq i$, and the loss of the masked data will not be calculated.

### 4.2 Internal Trend Mining Module

ITM module aims to complete the CFB feature. After KPM, we attained preliminary insufficient prediction $\hat{\mathbf{Y}}_{Init}^T$. In the ITM module, we first adopt a complete network to patch the mapping of CFB from the look-back window to the prediction window and then employ an adaptation network to fulfill the adaptive migration of data distribution from CFB to the label. The simplified framework is in Figure 3 and the specific framework is in Appendix C.2. We utilize another pair of encoder $\mathbb{R}^{L+P} \mapsto \mathbb{R}^D$ and decoder $\mathbb{R}^D \mapsto \mathbb{R}^{L+P}$ to learn the common embedding for entire known information. Unlike KPM, ITM takes all known information as input, i.e., data in the train and prediction window. Regarding the label, we pad preliminary predicted values $\hat{\mathbf{Y}}_{Init}^T$ at the prediction window: $\mathbf{Z}\mathbf{C}^T = Encoder(concat(\mathbf{C}_L^T, \mathbf{C}_P))$, $\mathbf{Z}\mathbf{Y}^T = Encoder(concat(\mathbf{Y}_L^T, \hat{\mathbf{Y}}_{Init}^T))$. To settle the forecast window puzzle, we use the complete network to extend known curves into unknown regions, which is designed as a linear network:

$$\mathbf{Z}\hat{\mathbf{C}}_{TP}^T = Complete\_network(\mathbf{Z}\mathbf{C}^T). \tag{5}$$

Additionally, despite there being a certain correlation between CFB and label, their distributions are not entirely identical. Based on this fact, we employ an adaptation network to adjust the label adaptively, with the ETG module (4.3) following closely behind. Since both the complete network and adaptation network operate on the hidden variable $\mathbf{Z}\mathbf{Y}^T$ based on the original attributes of labels, we merge them into the ITM module to distinguish it from the ETG module. After traversing from the ITM & ETG module, we acquire the desired latent variable $\mathbf{Z}\hat{\mathbf{Y}}_P^T$:

$$\mathbf{Z}\tilde{\mathbf{Y}} = Adaptation\_network(Complete\_network(\mathbf{Z}\mathbf{Y}^T)), \quad \mathbf{Z}\hat{\mathbf{Y}}_P^T = ETG\_module(\tilde{\mathbf{Z}\mathbf{Y}}). \tag{6}$$

Finally, the latent vector $\mathbf{Z}\hat{\mathbf{C}}_{TP}^T$ and $\mathbf{Z}\hat{\mathbf{Y}}_P^T$ are converted into target predicted values with decoder: $\hat{\mathbf{C}}_{TP}^T = Decoder(\mathbf{Z}\hat{\mathbf{C}}_{TP}^T)$, $\hat{Y}_P^T = Decoder(\mathbf{Z}\hat{\mathbf{Y}}_P^T)$. The final result is calculated by adding these two values with the reminder predicted values (acquired with the linear mapping of $\mathbf{Y}_L^R$, $\mathbf{C}_L^R$ in Figure 3, $\hat{\mathbf{C}}_{TP}^R = Linear(\mathbf{C}_L^R)$, $\hat{Y}_P^R = Linear(\mathbf{Y}_L^R)$):

$$\hat{\mathbf{C}}_{TP} = \hat{\mathbf{C}}_{TP}^T + \hat{\mathbf{C}}_{TP}^R, \quad \hat{\mathbf{Y}}_P = \hat{\mathbf{Y}}_P^T + \hat{\mathbf{Y}}_P^R. \tag{7}$$

To ensure the effectiveness of complete network, we introduce the prediction bias loss $\mathcal{L}_{be\_y}$:

$$\mathcal{L}_{be\_y} = \frac{2}{k^2} \sum_{i=0}^{k} \sum_{j=0}^{k} (\hat{\mathbf{C}}_{TP}[i,j] - \mathbf{C}_{TP}[i,j])^2. \tag{8}$$

### 4.3 External Trend Guide Module

**ETG** module is designed to settle the trend unobvious challenge, which uses aggregated spatial dimension to improve prediction results. Aggregated spatial dimension has stronger regularity, less noise, and is easier to estimate. Table 1 shows that the sample size of the hotel is approximately

$400k$, which is too extensive for direct model train. Therefore, referring to (Lu et al., 2022), we introduced a hierarchical sampling strategy to construct samples. This strategy assumes the global reconciliation matrix $\mathbf{P}$ is sparse, indicating that only child nodes belonging to the same parent node have calibration relationships with each other. This assumption aligns well with the geographic attributes of hotels. We aggregate hotels into business districts based on their geographic location and then to higher levels of urban granularity. First, we randomly pick a business district. Then, we sample $m$ hotels from this district, with the likelihood of selection increasing with each hotel's historical label value. These $m$ hotels' labels are combined to create a virtual parent node. Together, the $m + 1$ nodes form a sampled hierarchy that serves as model input. This hierarchical sampling maintains the sum constraint through virtual parent nodes, and label weighted sampling aligns the virtual sequence more with the real parent sequence. Moreover, these strategies reduce the reconciliation matrix's parameter count from $\mathcal{O}(n^2)$ to $\mathcal{O}(mn)$, significantly easing the model's computational load.

The framework of ETG is detailed in Figure 3 and Appendix C.3. The input of ETG is $\mathbf{Z\tilde{Y}}$ (Eq. (6)). $z_{i/j/k} = \mathbf{Z\tilde{Y}}[i/j/k]$ are the elements of $\mathbf{Z\tilde{Y}}$. According to the hierarchical sampling, we can adjust its shape from $[Bt, \cdots]$ to $[g, m, \cdots]$, where $Bt = g \times m$, indicating the number of original nodes, the number of nodes in high-level and in low-level. All subsequent actions are operated within the virtual hierarchical group. We obtain the key and query to calculate the reconciliation matrix $\mathbf{B}$ between nodes in the same hierarchical structure, where $\mathbf{B}(i, j)$ indicates the reconciliation relationship between the $i$th and $j$th nodes in the hierarchical structure:

$$e_{ij} = \mathbf{W}_q z_i \odot \mathbf{W}_k z_j, \quad \mathbf{B}[i, j] = \frac{exp(e_{ij})}{\sum_{k \in \mathcal{M}_i} exp(e_{ik})}, \tag{9}$$

where $\mathbf{W}_q \in \mathbb{R}^{d \times d}, \mathbf{W}_k \in \mathbb{R}^{d \times d}$ are the query and key parameter of model, $\mathcal{M}_i$ is the set of nodes in the current hierarchy of $i$. Then, we use the reconciliation matrix $\mathbf{B}$ to calibrate each sequence:

$$\hat{z}_i = \sum\nolimits_{k \in \mathcal{M}_i} \mathbf{B}[i, k] \times \mathbf{W}_v \times z_k, \tag{10}$$

where $\mathbf{W}_v \in \mathbb{R}^{d \times d}$ is the model's value parameter. Given the lower noise and greater regularity of parent nodes compared to child nodes, we refrain from applying representation calibration to parent nodes using masking techniques, aligning with the reconciliation process. During the model training process, we introduce reconciliation loss to implement hierarchical constraints:

$$\mathcal{L}_{recon} = \frac{1}{g^2} \sum\nolimits_{i=0}^{g} (\hat{\mathbf{Y}}_P[i^H] - \sum\nolimits_{j=0}^{m} \hat{\mathbf{Y}}_P[i, j])^2. \tag{11}$$

where $\hat{\mathbf{Y}}_P[i^H]$ is the estimated result of parent node. $\mathcal{L}_{recon}$ indicates the difference between the direct estimation of the parent node and sum of underlying estimation results, ensuring the sum of the calibrated estimated results of the underlying nodes approach parent node's estimated results.

### 4.4 ALGORITHM INFERENCE

To fully utilize demand information related to cross-future behavior, we constructed the demand-constrained loss. In practice, we have found that there are invisible boundaries on the amount of user demand. Inspired by (Avati et al., 2020; Gao et al., 2022), we are conscious that the boundary is informative for the model. Thus, we construct upper and lower limits based on the demand in transaction scenarios and design a demand-constrained loss to digest this boundary information. The principle is shown in Figure 4. From the perspective of likelihood estimation, we make assumptions about the distribution of labels when using the common MAE or MSE as a loss function. Taking the MSE loss function used in this paper as an example, the MSE loss function is based on the assumption that the predicted target follows a Gaussian distribution, and what our model infers is the mean of the Gaussian distribution. The actual observation may not necessarily be the mean of true Gaussian distribution. With the assistance of upper and lower limits, we can support the model hover the true distribution.

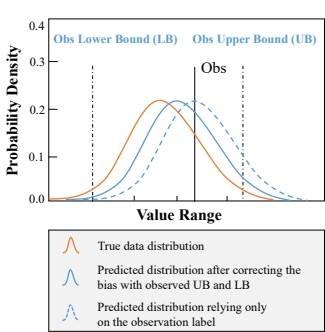

Figure 4: Demand-Constrained Loss.

In the check-in scenario of the hotel, we regard the truth value as label $y$ (i.e., $\mathbf{Y}$) and regard the value for the day of reservation as the lower demand bounds $y_l$, excluding entire CFB actions. We also consider the page view of the order page as the upper bound $y_u$, including the unconverted potential user data. The demand-constrained loss is as follows:

$$f_d(\hat{y}, y, y_l, y_u) = \begin{cases} (\hat{y} - y)^2 + \beta(\hat{y} - y_l)^2, & \text{if } \hat{y} < y_l \\ (\hat{y} - y)^2, & \text{if } y_l \leq \hat{y} \leq y_u \\ (\hat{y} - y)^2 + \beta(\hat{y} - y_u)^2, & \text{if } \hat{y_u} < \hat{y}, \end{cases} \quad (12)$$

where $\beta$ is hyperparameter. We define the main loss as $\mathcal{L}_y$, indicating the forecast bias of $\mathbf{Y}_P$:

$$\mathcal{L}_y = f_d(\hat{\mathbf{Y}}_P, \mathbf{Y}_P, \mathbf{Y}_P^L, \mathbf{Y}_P^U). \quad (13)$$

where $\mathbf{Y}_P^L$ is the lower bound matrix, and $\mathbf{Y}_P^U$ is the upper bound matrix. Through the aforementioned submodules, CRAFT enables the future trend of cross-future behavior to approach consummation and migrate it to the tendency cognition of label, aggregating to higher-level label to perceive more distinct and precise inclination subsequently. During the process, we obtain $\hat{\mathbf{C}}_P$ which is transferred from encoder to decoder to constrain the representation of koopman embedding, the intact matrix expression of prediction window of cross-future behavior $\hat{\mathbf{C}}_{TP}$, and ultimate desired prediction results $\hat{\mathbf{Y}}_P$. To achieve better model outcome, the loss of CRAFT consists of four parts, where $\alpha_n, n \in 1, 2, 3$ are hyper parameters used to balance multiple losses:

$$\mathcal{L} = \mathcal{L}_y + \alpha_1 \mathcal{L}_{be\_k} + \alpha_2 \mathcal{L}_{be\_y} + \alpha_3 \mathcal{L}_{recon}. \quad (14)$$

$\mathcal{L}_{be\_k}, \mathcal{L}_{be\_y}, \mathcal{L}_{recon}, \mathcal{L}_y$ are shown in Eq. (4), Eq. (8), Eq. (11), and Eq. (12) which corresponding to the recovery loss of $\hat{\mathbf{C}}_P$ in the KPM module, the prediction error of $\hat{\mathbf{C}}_{TP}$ in the ITM module, the reconstruction drift of $\hat{\mathbf{Y}}_P$ at the ETG module and the forecast deviation of label $\hat{\mathbf{Y}}_P$ respectively.

# 5 EXPERIMENTS

## 5.1 EXPERIMENTAL SETTINGS

### 5.1.1 DATASET

We conduct offline experiments on real-world dataset collected in May 2023 at ****. To reflect the model effects on different data distributions objectively, the prediction window we cover to verify the model's effectiveness contains both holidays and daily events. In addition, we set different forecast lengths $K \in \{7, 14, 30\}$, corresponding to lookback lengths $T \in \{30, 90, 180\}$. The dataset statistics are shown in Table 1. The hotels we use for verification are located in over 400 cities, covering more than 5000 business districts. The total sample size is around $400k$.

Table 1: Dataset Statistics.

| Hierarchy | Volume |
| --- | --- |
| # of city | 0.4k |
| # of business | 5k |
| # of hotel | 400k |

### 5.1.2 BASELINE METHODS AND EVALUATION METRICS

The baseline methods for comparison include MQ-RNN (Wen et al., 2017), Informer (Zhou et al., 2021), DLinear (Zeng et al., 2023), Koopa (Liu et al., 2024), TFT (Lim et al., 2021), Autoformer (Wu et al., 2021) and Fedformer (Zhou et al., 2022).

Weighted Mean Absolute Percentage Error ($wMAPE$) is adopted to measure the models' performance in offline experiments:

$$wMAPE = \frac{\sum |y - \hat{y}|}{\sum y}, \quad (15)$$

where $y$ denotes the ground truth and $\hat{y}$ denotes the predicted value. In hotel booking situations, the data distribution is a significant imbalance, adopting $wMAPE$ as a performance metric can effectively alleviate zero values issues. In addition, $wMAPE$ allows assigning different weights to different ground truth, thus increasing the evaluation robustness. In addition, $MAE$ and $RMSE$, two widely used metrics in time series forecasting, are adopted to evaluate the models' performance.

Table 2: Comparative forecasting results with ~~the look-back window length of $L \in \{30, 90, 180\}$ and~~ prediction window length of $P \in \{7, 14, 30\}$ respectively, correspond one-to-one with the look back window $L \in \{30, 90, 180\}$. The unit of length is days. The best results are highlighted in bold and the second best results are highlighted with a underline.

| Model | Length $P$ | Only label | | | With CFB as covariate | | |
|---|---|---|---|---|---|---|---|
| | | MAE | RMSE | wMAPE | MAE | RMSE | wMAPE |
| Autoformer | 7 days | 0.9319 | 2.9374 | 0.7030 | 0.9631 | 2.9132 | 0.7265 |
| | 14 days | 1.0017 | 2.3737 | 0.9830 | 0.9799 | 2.3566 | 0.9624 |
| | 30 days | 0.9658 | 2.2737 | 0.9910 | 1.2101 | 2.7625 | 1.2414 |
| Fedformer | 7 days | 0.9280 | 2.9147 | 0.7000 | 0.9191 | 2.9100 | 0.6933 |
| | 14 days | 0.8744 | 2.3333 | 0.8605 | 0.9071 | 2.4167 | 0.8923 |
| | 30 days | 0.9109 | 2.1718 | 0.9347 | 1.6507 | 4.7499 | 1.6933 |
| TFT | 7 days | 0.9584 | 2.9184 | 0.7195 | 0.9535 | 2.9043 | 0.7215 |
| | 14 days | 0.8643 | 2.2745 | 0.8535 | 0.8524 | 2.2730 | 0.8468 |
| | 30 days | 0.8294 | 2.3346 | 0.8494 | 0.8301 | 2.2945 | 0.8693 |
| DLinear | 7 days | 0.9825 | 2.9539 | 0.7412 | 0.9822 | 2.9566 | 0.7410 |
| | 14 days | 0.8555 | 2.3279 | 0.8418 | 0.8571 | 2.3572 | 0.8427 |
| | 30 days | 0.8125 | **1.9499** | 0.8337 | 0.8089 | 1.9634 | 0.8298 |
| Informer | 7 days | 0.9731 | 2.9251 | 0.7341 | 0.9365 | 2.9169 | 0.7065 |
| | 14 days | 0.8034 | **2.2653** | 0.7906 | 0.8203 | 2.2969 | 0.8065 |
| | 30 days | 0.7699 | 1.9618 | 0.7899 | 0.8063 | 1.9844 | 0.8271 |
| MQ-RNN | 7 days | 0.9007 | 3.0013 | 0.6895 | 0.9064 | 3.0185 | 0.6830 |
| | 14 days | 0.7403 | 2.5217 | 0.7478 | 0.7415 | 2.4554 | 0.7381 |
| | 30 days | 0.6958 | 2.1756 | 0.7142 | 0.7029 | 2.2161 | 0.7215 |
| Koopa | 7 days | 0.9024 | 2.9047 | 0.6943 | 0.8927 | 2.8948 | 0.6818 |
| | 14 days | 0.7440 | 2.3485 | 0.7326 | 0.7475 | 2.4327 | 0.7350 |
| | 30 days | 0.7045 | 2.1943 | 0.7276 | 0.6984 | 2.3456 | 0.7176 |
| **CRAFT** | 7 days | | | | **0.8480** | **2.8654** | **0.6706** |
| | 14 days | | | | **0.7237** | 2.2696 | **0.7121** |
| | 30 days | | | | **0.6895** | 2.0121 | **0.7078** |

### 5.1.3 IMPLEMENTATION

All experiments are implemented with Python 3.8.5 and Pytorch 1.12.1 We conduct them on the cloud servers with two NVIDIA Tesla T4 GPUs with 16GB VRAM each. We initialize the network parameters with *Xavier Initialization* (Glorot & Bengio, 2010). Each parameter is sampled from $N(0, \mu^2)$, where $\mu = -\sqrt{2/(n_{in} + n_{out})}$. $n_{in}, n_{out}$ denote the number of input and output neurons, respectively. In actuality, ~~the kernel size of the moving average in temporal decomposition is 15,~~ the $\lambda$ of ridge regression for solving the koopman matrix in the ITM module is 0.1, the number of child nodes $m$ at the virtual hierarchy is set as 15 during hierarchical sampling. In addition, we train all models by setting the mini-batch size to 256 and using the Adam optimizer with a learning rate of 0.001. Except for MQRNN with the quantile loss at 0.5, all other models choose MSE as the training loss. The number of training epochs is 2 on the dataset, and the value of each experimental result is the average of 5 repeated tests. The detailed model configuration selections for all models are provided in Appendix F.

## 5.2 OFFLINE EXPERIMENTS

### 5.2.1 COMPARISON WITH BASELINES

The comparative results are shown in Table 2. For fairness, we compared both the original baseline methods and the improved versions with CFB. CRAFT achieves the best performance, significantly outperforming the other baselines. We obtain the following observations from Table 2:

- Compared to the original model, directly integrating CFB into the existing framework does not yield significant performance enhancements. In some cases, it even leads to performance degradation. The experimental results confirm that, despite its indispensable role, effectively applying CFB presents considerable challenges.
- Compared with the optimal baseline, CRAFT improves by at least $\{0.0447, 0.0166, 0.0063\}$ and $\{0.0112, 0.0205, 0.0064\}$ in $MAE$ and $wMAPE$ metrics in the prediction window of $\{7,$

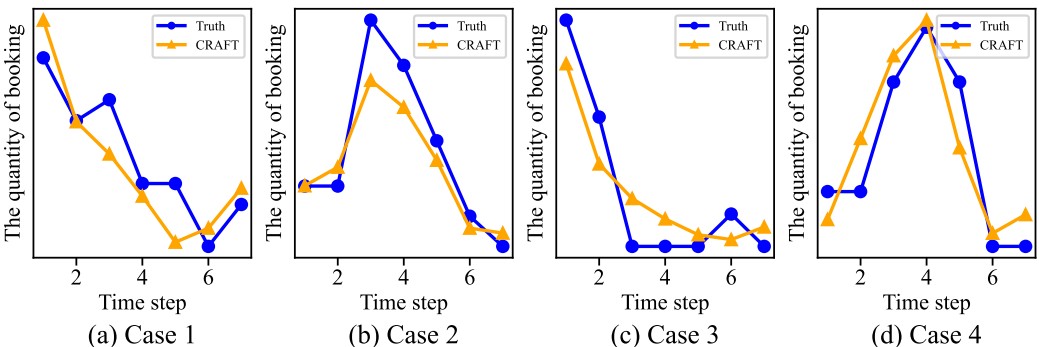

Figure 5: Case study on (a)-(d) four different cases, where blue line is the truth value and the orange line is the prediction of CRAFT.

Table 3: Ablation study of CRAFT.

| KPM | ITM | ETG | Demand Loss | $MAE$ | $RMSE$ | $wMape$ |
|:---:|:---:|:---:|:---:|:---:|:---:|:---:|
| ✓ | × | × | × | 0.9440 | 3.1507 | 0.7122 |
| ✓ | ✓ | × | × | 0.9090 | 2.9416 | 0.6857 |
| ✓ | ✓ | ✓ | × | 0.8557 | 2.7555 | 0.6455 |
| ✓ | ✓ | ✓ | ✓ | 0.8480 | 2.7480 | 0.6397 |

14, 30}. In $RMSE$ metric, CRAFT ranks the best and the second best when the length of prediction window is 7 and 14. To sum up, CRAFT performs better than baseline models in various prediction lengths, demonstrating stable superiority. The reason behind the excellent results is that CRAFT utilizes the KPM and the ITM module to fully explore the CFB, adopts the ETG to transfer the trend of high-level time series to low-level time series, and employs demand-constrained loss to correct the prediction distribution deviation.

- Experimental results indicate that CRAFT achieves the best results at the prediction length of 7. The reason behind this phenomena is that when the prediction length increases, the sparsity and unknown properties of the CFB become more obvious.
- In the experimental results, the longer the prediction window, the smaller the prediction error. The reason is that the prediction windows consist of daily data and holiday data, and holiday patterns are more difficult to predict. As the length of the prediction window increases, the proportion of holiday data decreases and the overall prediction error reduces.

### 5.2.2 ABLATION STUDY

To verify the effectiveness of each module in CRAFT, we conduct an ablation study on the setting of the prediction window's length is 7, as CRAFT achieves the best results on this condition. The experiment results of the ablation study are listed in Table 3. According to Table 3, we can know that the ITM, ETG module, and the constrain-demand loss all have positive impact on improving the performance of CRAFT. Among them, ITM and ETG module achieve the most obvious improvement with $0.035, 0.053$ on $MAE$, $0.2091, 0.1861$ on $RMSE$, $0.0265, 0.0402$ on $wMAPE$.

### 5.2.3 CASE STUDY

To verify that the model is effective in capturing future trends, we selected samples from the dataset with different trends for validation. As shown in Figure 5, the trend varies from sample to sample event for the same event impact: some hotels reached their peak in the early stages of the holiday and showed an overall downward trend (Figure 5 (a), (c)); some hotels had high traffic in the middle holiday period and showed an overall mountain shape (Figure 5 (b), (d)). The predicted trend of CRAFT is also not constant but changes with the actual trend of the sample, which indicates the reliability of CRAFT.

Table 4: Online A/B test result during holiday on IWR and PHDI metric.

| Holiday | IWR[*] | | PHDI[†] | |
|---|---|---|---|---|
| | MQ-RNN | CRAFT | MQ-RNN | CRAFT |
| 2023 Mid-autumn | 0.0513 | 0.0306 | 0.3659 | 0.2983 |
| 2023 National Day | 0.0534 | 0.0311 | 0.3694 | 0.2990 |
| 2024 New Year's Day | 0.0601 | 0.0354 | 0.3661 | 0.3093 |
| 2024 Spring Festival | 0.0583 | 0.0337 | 0.3655 | 0.3047 |

[*] IWR means inventory waste rate. † PHDI means the proportion of hotels with depleted inventory.

### 5.3 ONLINE A/B TEST

To further verify the performance of CRAFT in the real online environment, we apply CRAFT to holiday inventory negotiations. In the real application, we need to predict the hotel sales before holidays, and business developers (BD) will check whether the inventory is sufficient based on the prediction results of our model. If not, they will negotiate with the hotel in advance based on the prediction results to give more inventory. We select the MQ-RNN as the baseline and utilize $IWR$ and $PHDI$ metrics to measure the overall impact of different models. The definition of $IWR$ and $PHDI$ refers to Appendix B.4. $IWR$ and $PHDI$ are defined according to specific scenarios and are the most concerned metrics in BD negotiations. Moreover, as we cannot equally assign daily traffic to each model, such as testing personalized recommendation systems, we randomly divided the hotels into two groups for the MQ-RNN and CRAFT models.

For both the $IWR$ and $PHDI$ metrics, the smaller the value, the better the model performance. The online A/B test results are shown in Table 4. Compared with MQ-RNN, CRFAT achieves significant improvement on these two metrics. Indeed, based on the data from four holidays, CRAFT method has an average improvement of $0.0231$ and an improvement rate of $41.35\%$ on the $IWR$ metric. On the $PHDI$ metric, the improvement value and improvement rate are $0.0639$ and $17.42\%$, respectively. The results above illustrate the effectiveness of CRAFT in the real application.

## 6 CONCLUSION

In this paper, inspired by real-world application, we define **Cross-Future Behavior (CFB)**. CFB is a kind of features that occur before the current time but take effect in the future, containing true information related to future events. For the application of CFB in the time series forecasting problem, we propose an improved method, named **Cross-Future Behavior Awareness based Time Series Forecasting method (CRAFT)**. CRAFT regards the trend of CFB as prior information to predict the trend of target time series data. Specifically, CRAFT is composed of three main modules, the Koopman Predictor Module (KPM), the Internal Trend Mining Module (ITM), and the External Trend Guide Module (ETG). The first two modules aim to mine the prediction trends from partial CFB, and the ETG module aims to acquire more representative trends from higher levels. In addition, CRAFT adopts demand-constrained loss to correct the distribution of prediction results. We conduct experiments on real-world dataset. Experiments on both offline and online tests demonstrate the effectiveness of CRAFT.

This paper only explores the application of CFB and CRAFT on the e-commerce area. Electricity demand, stock price, and disease spread forecasting typically do not have clear lead-up operation events, making it difficult to apply our method. In addition, the current public available dataset (e.g., ETT(Zhou et al., 2021), ECL[1], Weather[2]) does not contain the CFB feature or similar feature, we only verify CRAFT on our dataset limited by this actual situation. In the future, we will further collect related data to form a series benchmark dataset. Moreover, we will continue to generalize the definition of CFB to enable CRAFT method to be applied more broadly.

---

[1] ECL dataset was acquired at https://archive.ics.uci.edu/ml/ datasets/ElectricityLoadDiagrams20112014.

[2] Weather dataset was acquired at https://www.ncei.noaa.gov/ data/local-climatological-data/.

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

# APPENDIX

## A  SUPPLEMENT INTRODUCTION TO CROSS-FUTURE BEHAVIOR

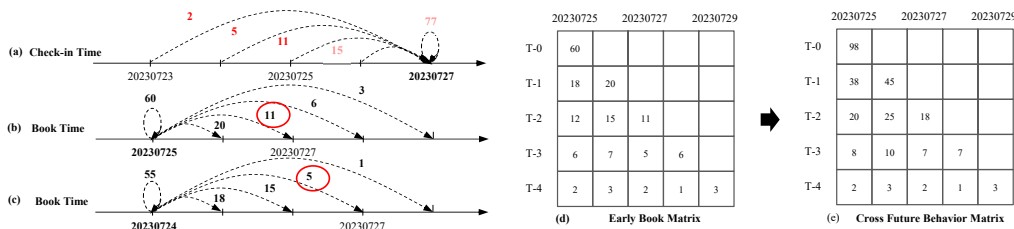

Figure 6: Cross-Future Behavior (CFB) on hotels booking example.

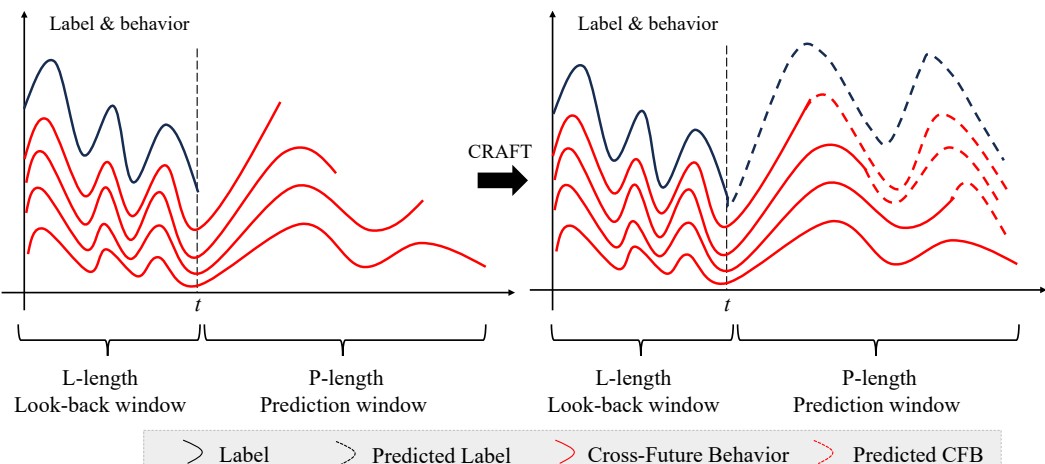

Figure 7: An illustration of label and Cross-Future Behavior (CFB). The top black line is the label, the red lines are CFB, the dash black line is the predicted lable, and the dashed red lines are the unknown part of CFB. Left: the original inputs of the model, consisting of $L$-length look-back window and $P$-length prediction window. Right: the model's outputs to be predicted, forming a complete prediction window.

**Cross-Future Behavior (CFB)** is feature that occurs before the current time but takes effect in the future. Taking hotels as an example to illustrate CFB, as shown in Figure 6. A user can book a hotel room on that day or make an early book for a future date (Figure 6 (b), (c)), a hotel's check-in rooms within a day originate from bookings made on that day and previous days (Figure 6 (a)). Assuming today is July 25, the bookings made from July 23 to July 25 for July 27 (2 rooms, 5 rooms, 11 rooms) exemplify the early book matrix (Figure 6 (d)). Furthermore, in order to obtain more comprehensive information, we accumulate the quantities of early books to obtain the cross-future matrix (Figure 6 (e)). The early book matrix and cross-future behavior matrix decompose the number of hotel check-in rooms for a given day based on the book dimension. Hotel examples will also be used later in the text for ease of understanding and will not be repeated below. As shown in Figure 7, the left graph shows all information known at the moment of predicting $t$, and after prediction by the model, the right graph completes the unknown information to be predicted.

However, the use of CFB in TSF problem faces two main challenges: **1) CFB is sparse and partial.** Consumers can book items at any time, causing CFB to remain fully observed until the last minute. As shown in Figure A(d), half of the data in the cross-future matrix cannot be obtained. When making predictions, the data for row T-0 can only be fully obtained at the end of the day, meaning only rows T-1 to T-n are available for forecasting, showing that CFB is sparser and smaller than

other features such as hotel historical sales. Thus, if CFB is simply incorporated into the model, the prediction model may be unable to apply CFB features correctly and even make incorrect predictions due to CFB. **2) The trend of CFB is unobvious.** As shown in Figure 2, compared with the sales trend in a high-level business district, the sales trend in an individual hotel is unobvious. CFB has the same nature, and the trend in an individual hotel is much unobvious compared with that in a high-level business district. CFB can reflect future trends to a certain extent, however, the coverage of CFB for some hotels may be too atypical to reveal a trend. Sales at higher-level districts are much more representative than in individual hotels. If there is a rising trend at a higher level, it is also possible that the sales of individual hotels may increase. How to leverage the sales trends of the higher level to guide the trends exploration for individual hotels is the second challenge.

# B    SUPPLEMENT PRELIMINARIES

## B.1    PROBLEM DEFINITION

Table 5: Important notations used in this paper.

| Notation | Definition |
|---|---|
| $t$ | the time point |
| $L$ | the length of look-back window |
| $P$ | the length of prediction window |
| $\mathbf{Y}_L$ | the time series in look-back window |
| $\mathbf{Y}_P$ | the time series (i.e., predicted output) in prediction window |
| $\mathbf{Y}_L^T$ | the trend part of $\mathbf{Y}_L$ in look-back window |
| $\mathbf{Y}_L^R$ | the reminder part of $\mathbf{Y}_L$ in look-back window |
| $\mathbf{Y}_P^U$ | upper limit of $\mathbf{Y}_P$ |
| $\mathbf{Y}_P^L$ | lower limit of $\mathbf{Y}_P$ |
| $\mathbf{C}_t$ | Cross-Future Behavior (CFB) |
| $\mathbf{C}_L$ | CFB in look-back window |
| $\mathbf{C}_{TL}$ | the ground truth of CFB in look-back window |
| $\mathbf{C}_P$ | CFB in prediction window |
| $\mathbf{C}_L^T$ | trend part of $\mathbf{C}_L$ |
| $\mathbf{C}_L^R$ | reminder part of $\mathbf{C}_L$ |
| $\mathbf{X}_t$ | covariate features |

For clarity, we summarize the important notations used in this paper as Table 5.

## B.2    KOOPMAN THEORY

Koopman Theory (Koopman, 1931) mainly focuses on the dynamical system, studying the evolution of state variables along one or more coordinate axes (usually time). Koopman theory transforms finite-dimensional nonlinear dynamics problems into infinite-dimensional linear dynamics problems, by projecting the state into the space of measurement function $g(y)$ and evolving forward by an infinite-dimensional linear operator $\mathcal{K}$ which is also called the Koopman matrix. For a discrete-time dynamical system, it can be transformed as:

$$\begin{aligned} y_{t+1} &= F(y_t), \\ \mathcal{K}(g(y_k)) &= g(F(y_k)) = g(y_{k+1}), \end{aligned} \quad (16)$$

where $F$ is the evolution function in dynamic systems. In practice, we apply a finite linear matrix $K$ to approach the infinite value. Recently, there have been several studies applying koopman theory to the filed of time series prediction. Taking inspiration from (Liu et al., 2024), we treat the prediction

window as a time period and calculate the koopman matrix between time periods. The Koopman matrix can be approximated by ridge regression, i.e.,:

$$B = KA \Rightarrow K = (A^T A + \lambda E)^{-1} A^T B, \tag{17}$$

where $\lambda$ is the ridge coefficient to command the weight of regularization terms, $E$ is the identity matrix. Ridge regression effectively prevents model over-fitting and solves multicollinearity problems under Deficient-rank conditions by introducing L2 regularization terms.

### B.3 HIERARCHICAL STRUCTURE

Multivariate time series data typically have a hierarchical structure, where each upper-level time series is computable by summing over the appropriate lower-level time series. Hierarchical prognostics need to satisfy aggregation constraints, i.e., they need to ensure that the sum of predictions of some parent node and the sum of the predictions of its child nodes are approximately equal. Reconciliation-based hierarchical prognostics approach is currently the state-of-the-art solution in hierarchical time series estimation, which is based on the estimated values of existing nodes, and the calibrated estimated results are obtained through the following formula:

$$\tilde{y}_t = \mathbf{SB}\hat{y}_t. \tag{18}$$

Where $\mathbf{S}$ is the hierarchical structure matrix, the value between node pairs with parent-child relationships is 1, otherwise it is 0. $\mathbf{B}$ is the reconciliation matrix which represents the calibration relationship between the bottom nodes and is the parameter to be solved.

### B.4 DEFINITION OF $IWR$ AND $PHDI$ METRIC

In practical application scenarios, in addition to the negotiated inventory based on predict results, sellers usually provide some additional inventory for unexpected needs. Therefore, there may be situations where the label value is greater than the predict result. Thus, the $IWR$ and $PHDI$ indicators of trade-off are adopted online to evaluate the quality of prediction performance at ****. we define the truth value as label $Y_P$, and define the predict result as $\hat{Y}_P$ as stated in the main text, m is the number of samples. IWR means inventory waste rate. If the predicted value is on the high side, there will be some inventory that has not been consumed, which is a wasted resource. namely,

$$IWR = \frac{1}{m} \begin{cases} 0, & \text{if } \hat{Y}_P[i] \leq Y_P[i] + b \\ \frac{\hat{Y}_P[i] - Y_P[i]}{\hat{Y}_P[i]}, & \text{if } \hat{Y}_P[i] > Y_P[i] + b, \end{cases} \tag{19}$$

where b is the buffer amount, which is the acceptable error range for BD. we set $b = 1$. The more predicted values are higher than the true values, the larger the IWR is. $PHDI$ means the proportion of hotels with depleted inventory, which is defined as follows:

$$PHDI = \frac{1}{m} \begin{cases} 1, & \text{if } \hat{Y}_P[i] < Y_P[i] - b \\ 0, & \text{if } \hat{Y}_P[i] \geq Y_P[i] - b. \end{cases} \tag{20}$$

If our predicted inventory is depleted, it indicates that our prediction of user demand is underestimated. The more hotels with predicted values lower than the true values plus buffer, the larger the PHDI is.

## C   Supplement Method Architecture

### C.1   Koopman Predictor Module

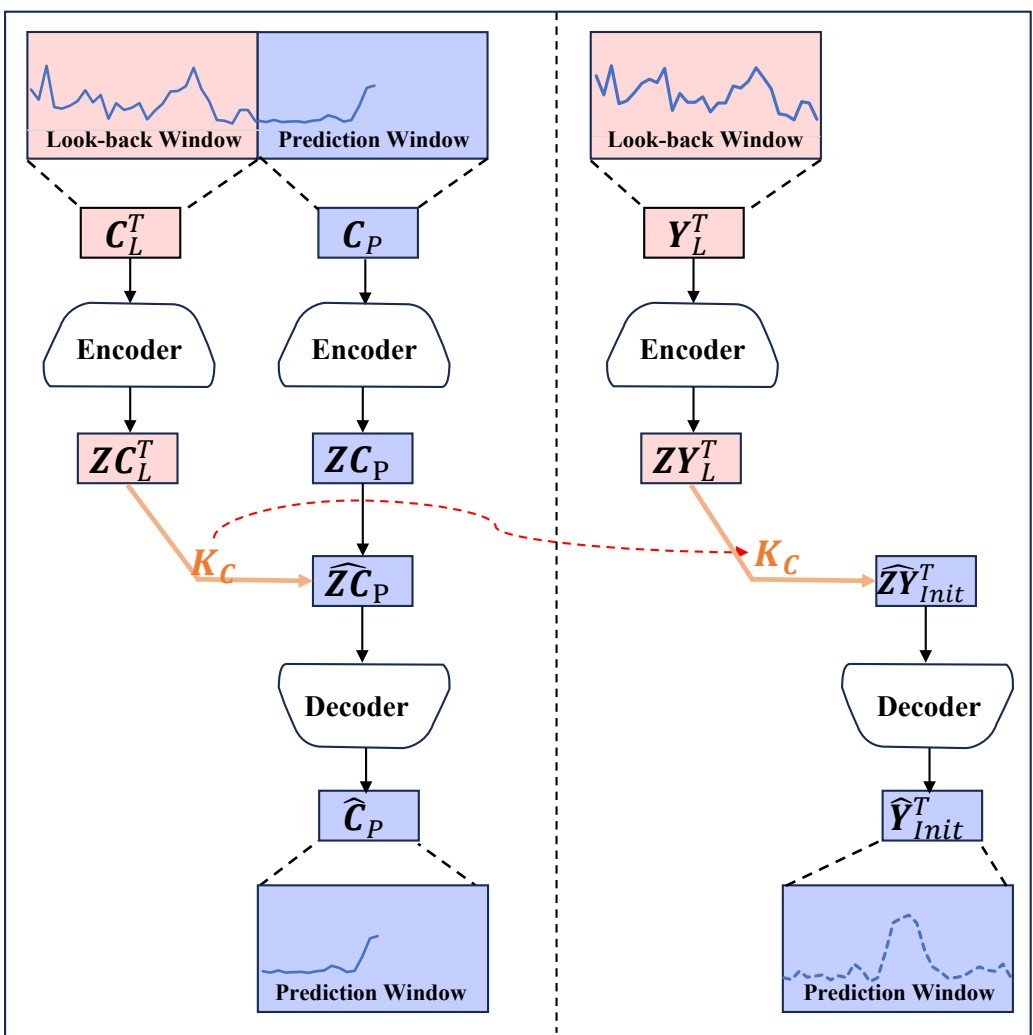

Figure 8: Framework of Koopman Predictor Module (KPM). The red portion indicates the look-back window and the purple portion is the prediction window. The left side is Cross-Future Behavior-related operations while the right side is label-related operations. In addition, the encoder and decoder are shared by both, the $\mathbf{K}_C$ is computed by the Cross-Future Behavior side and applied to the label side.

Figure 8 is the framework of Koopman Predictor Module (KPM).

## C.2 INTERNAL TREND MINING MODULE

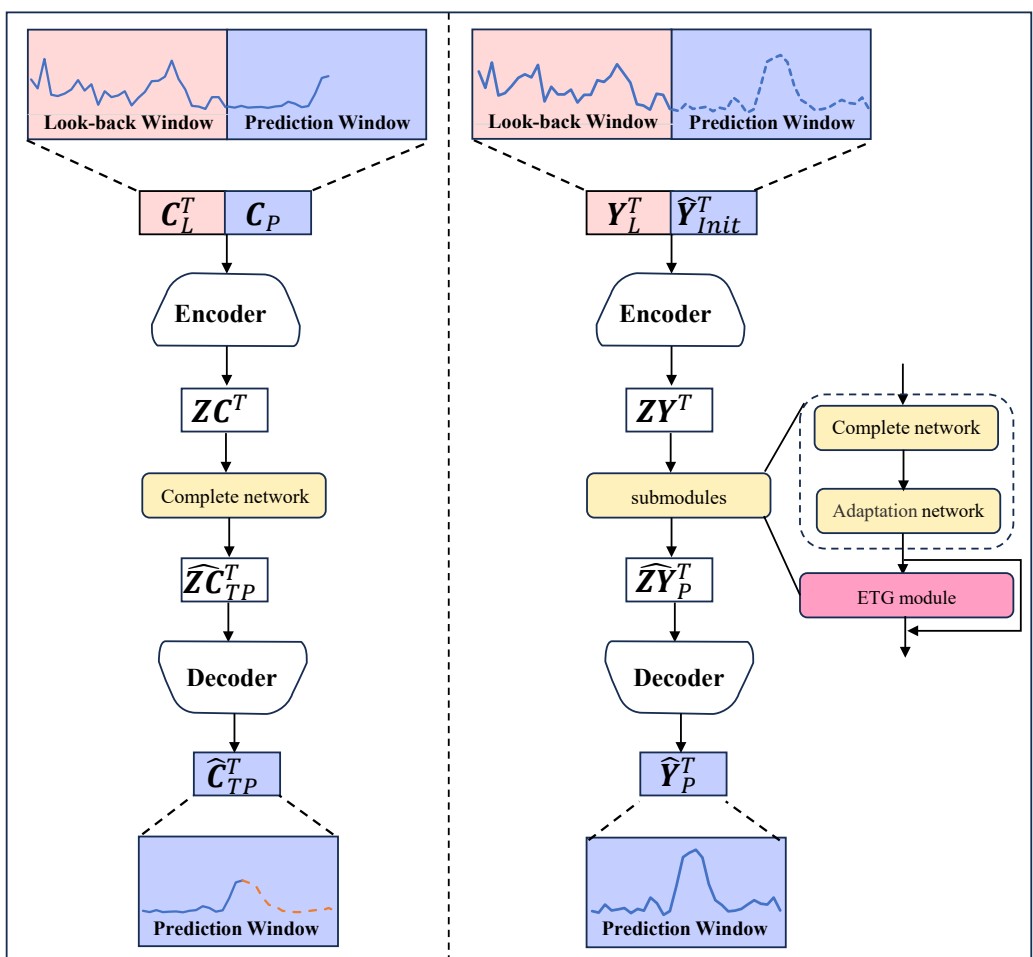

Figure 9: The framework of Internal Trend Mining Module (ITM). The meanings of red and purple portions are consistent with that in Figure 8. The Encoder, Decoder, and Complete network are all shared between the left and right sides.

Figure 9 is the framework of Internal Trend Mining Module (ITM).

## C.3 EXTERNAL GUIDE MODULE

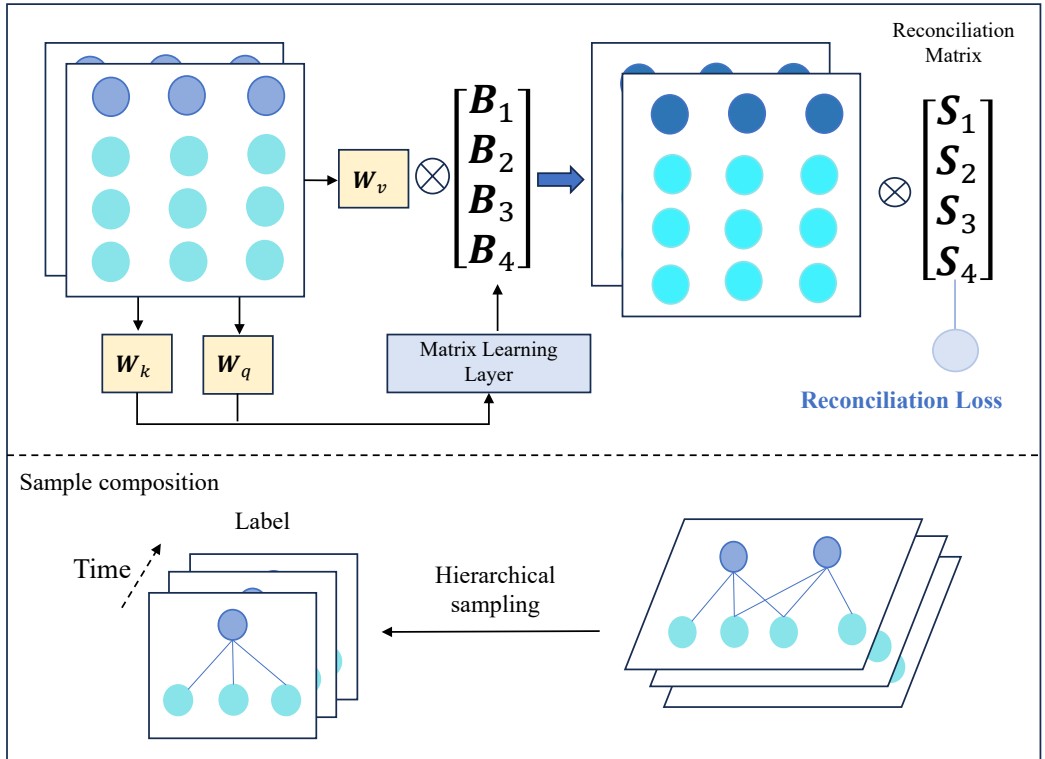

Figure 10: The framework of External Trend Guide Module (ETG). Top: The external trend guide module embedded in Figure 9, accompanied by reconciliation loss. Bottom: The sample composition matched with the ETG module which adopts a hierarchical sampling.

Figure 10 is the framework of External Trend Guide Module (ETG).

# D  THEORETICAL ANALYSIS

## D.1  THE CORRELATION ANALYSIS BETWEEN $\mathbf{K}_{TC}$ AND $\mathbf{K}_C$

Assuming we know the complete value $\mathbf{C}_{TP}$ of CFB, the logic for calculating the value of koopman matrix $\mathbf{K}_{TC}$ is as follows:

$$Encoder(\mathbf{C}_L^T) = \mathbf{Z}\mathbf{C}_L^T, Encoder(\mathbf{C}_{TP}) = \mathbf{Z}\mathbf{C}_{TP}. \tag{21a}$$

$$\mathbf{Z}\mathbf{C}_L^T \times \mathbf{K}_{TC} = \mathbf{Z}\mathbf{C}_{TP}, \tag{21b}$$

inwhich $\mathbf{K}_{TC}$ contains the trend information carried by CFB that we intend to learn.

In practice, the value we can acquire is $\mathbf{C}_{TP} = \mathbf{C}_P \odot L$, where $\odot$ represents the hadamard product and $\mathbf{L} = \begin{bmatrix} 1 & \cdots & 0 \\ \vdots & \ddots & \vdots \\ 1 & \cdots & 1 \end{bmatrix}$ is the unit lower triangular matrix. Therefore, in this section, we will demonstrate the relationship between $\mathbf{K}_{TC}$ and $\mathbf{K}_C$ utlized in the model, and further verify the rationality and necessity of KPM and ITM in the CRAFT.

In KPM module, the calculation process used for $\mathbf{K}_C$ value is as follows:

$$Encoder(\mathbf{C}_L^T) = \mathbf{Z}\mathbf{C}_L^T, Encoder(\mathbf{C}_P) = \mathbf{Z}\mathbf{C}_P. \tag{22a}$$

$$\mathbf{Z}\mathbf{C}_L^T \times \mathbf{K}_C = \mathbf{Z}\mathbf{C}_P, \tag{22b}$$

The encoder operation is a MLP, which can be defined as

$$Encoder() := tanh(\mathbf{W}\mathbf{X} + \mathbf{b}). \tag{23}$$

From equation 21a, 22a and 23, it can be concluded that

$$\mathbf{Z}\mathbf{C}_{TP} = tanh(\mathbf{W}\mathbf{C}_{TP} + \mathbf{b}), \tag{24}$$

$$\begin{aligned} \mathbf{Z}\mathbf{C}_P &= tanh(\mathbf{W}\mathbf{C}_P + \mathbf{b}), \\ &= tanh(\mathbf{W}(\mathbf{C}_{TP} \odot \mathbf{L})) + \mathbf{b}. \end{aligned} \tag{25}$$

From equation 21b and 24,

$$\begin{aligned} \mathbf{K}_{TC} &= \mathbf{Z}\mathbf{C}_L^{T^{-1}}\mathbf{Z}\mathbf{C}_{TP} \\ &= \mathbf{Z}\mathbf{C}_L^{T^{-1}} tanh(\mathbf{W}\mathbf{C}_{TP} + \mathbf{b}) \end{aligned} \tag{26}$$

can be derived. Similarly, from 22b and 25, it can be concluded that

$$\begin{aligned} \mathbf{K}_C &= \mathbf{Z}\mathbf{C}_L^{T^{-1}}\mathbf{Z}\mathbf{C}_P \\ &= \mathbf{Z}\mathbf{C}_L^{T^{-1}} tanh(\mathbf{W}(\mathbf{C}_{TP} \odot \mathbf{L}) + \mathbf{b}) \\ &= \mathbf{Z}\mathbf{C}_L^{T^{-1}} tanh(\mathbf{W}(\mathbf{C}_{TP} \odot (\mathbf{L} + \mathbf{1} - \mathbf{1})) + \mathbf{b}) \\ &= \mathbf{Z}\mathbf{C}_L^{T^{-1}} tanh(\mathbf{W}(\mathbf{C}_{TP} \odot \mathbf{1} - \mathbf{C}_{TP} \odot (\mathbf{1} - \mathbf{L})) + \mathbf{b}) \\ &= \mathbf{Z}\mathbf{C}_L^{T^{-1}} tanh(\mathbf{W}(\mathbf{C}_{TP} + \mathbf{b}) + (-\mathbf{W}(\mathbf{C}_{TP} \odot (\mathbf{1} - \mathbf{L})))) \end{aligned} \tag{27}$$

If we define $x : \mathbf{W}(\mathbf{C}_{TP} + \mathbf{b}), y := -\mathbf{W}(\mathbf{C}_{TP} \odot (\mathbf{1} - \mathbf{L}))$, then the above equation can be derived as

$$\begin{aligned} \mathbf{K}_C &= \mathbf{Z}\mathbf{C}_L^{T^{-1}} tanh(x + y) \\ &= \mathbf{Z}\mathbf{C}_L^{T^{-1}} \frac{tanh(x) + tanh(y)}{1 + tanh(x)tanh(y)} \\ &= \mathbf{Z}\mathbf{C}_L^{T^{-1}} tanh(x) \frac{1 + \frac{tanh(x)}{tanh(y)}}{1 + tanh(x)tanh(y)} \\ &= \mathbf{K}_{TC} \frac{1 + \frac{tanh(x)}{tanh(y)}}{1 + tanh(x)tanh(y)}. \end{aligned} \tag{28}$$

By the same token, it can be calculated that

$$\mathbf{ZC}_P = \mathbf{ZC}_{TP} \frac{1 + \frac{tanh(x)}{tanh(y)}}{1 + tanh(x)tanh(y)}. \tag{29}$$

Based on the above reasoning, there is a certain correlation between $\mathbf{K}_{TC}$ and $\mathbf{K}_C$, and the correlation will be fitted through the subsele.

### D.2 THE DISTRIBUTION CONSISTENCY VERIFICATION OF CFB AND LABEL

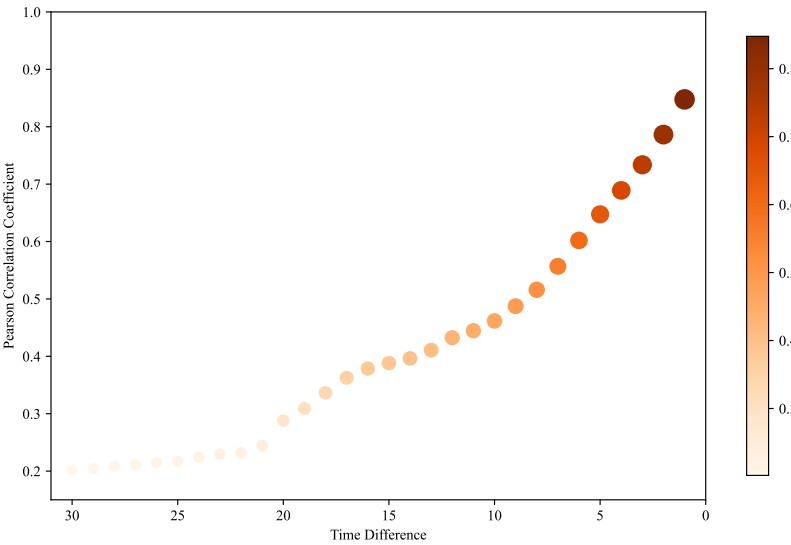

Figure 11: Pearson correlation between the CFB and label.

We verify the consistency between CFB and label from the data distribution view. Specifically, we randomly select one month of training data and calculate pearson correlation between label in look-back window and CFB in $p$-length prediction window. The calculation results are shown in Figure 11, where the $x$-axis is the length $p$ of prediction window and $y$-axis is the calculated pearson correlation. According to Figure 11, we can know that there are strong consistency between the the CFB and label. Especially, the smaller the window length $p$, the greater the correlation between CFB and label. This phenomena is also consistent with the experimental results in Table 2, that is, the smaller the prediction window length, the better the experimental results.

# E EXPERIMENTAL RESULTS

## E.1 VISUALIZATION OF KEY MODULES

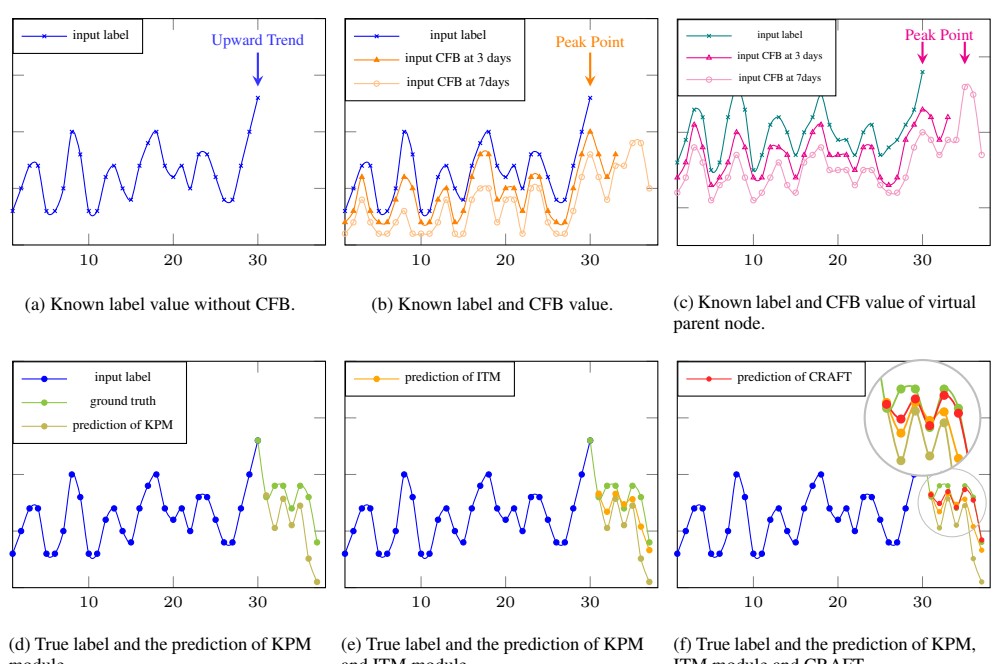

(a) Known label value without CFB.

(b) Known label and CFB value.

(c) Known label and CFB value of virtual parent node.

(d) True label and the prediction of KPM module.

(e) True label and the prediction of KPM and ITM module.

(f) True label and the prediction of KPM, ITM module and CRAFT.

Figure 12: Visualization of key modules using data of hotel A.

Figure 12 visualizes the original input data of CRAFT method sub-figure(a)-(c) and the prediction results after different modules sub-figure(d)-(f). According to Figure 12(a)-(c), we can know that there are strong consistency between the CFB and input label. When the prediction window is smaller, the consistency between CFB and label is stronger. This phenomena is also verified with results in Table 2 and Figure 11. Further, as shown in Figure 12(c), compared with CFB in the original node, CFB value of the virtual parent node tends towards the trend of the label more.

Figure 12(d)-(f) show the phased prediction results after KPM, ITM, and all modules of CRAFT, respectively. According to these three sub-figures, we can observe that these phased prediction results are closer to the ground truth label step by step. This is consistent with the results in Table 3, demonstrating the effectiveness of each module in the CRAFT method.

Figure 13: Hyper parameter analysis

Figure 14: Time complexity analysis

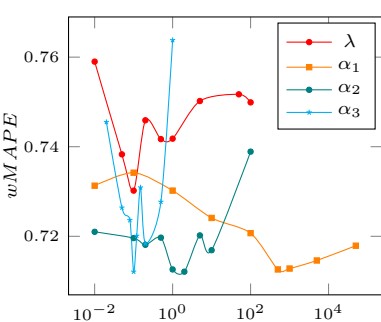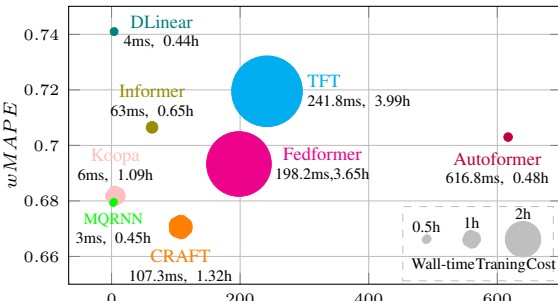

### E.2 HYPER PARAMETER ANALYSIS

Figure 13 analyzes the effects of hyper parameter $\lambda, \alpha_1, \alpha_2, \alpha_3$ to the $wMAPE$ metric. Firstly, all of these hyper paramter cover the interval from $10^{-2}$ to $10^2$. Based on the performance of the model, it will further explore areas with better property. In the end, $\lambda$ varies in $[10^{-2}, 10^2]$, $\alpha_1$ varies in $[10^{-2}, 5 * 10^4]$, $\alpha_2$ varies in $[10^{-2}, 10^2]$, and $\alpha_3$ varies in $[10^{-2}, 10^0]$. According to Figure 13, we can know that the value of hyper parameters affects the performance of the model to a certain extent, and the selection of model parameters is very important.

### E.3 TIME COMPLEXITY ANALYSIS

Figure 14 shows the time complexity of different methods. The $x$-axis is the inference cost for a batch of data, the $y$-axis is the performance on $wMAPE$ metric, and the circle size represents the wall-time training cost. From Figure 14, we can know that the training and inference costs of the proposed CRAFT method are both moderate among all baseline methods, faster than other most Transformer based methods. In addition, CRAFT outperforms all baseline methods in terms of $wMAPE$ performance.

### E.4 USING CFB AS LABEL

Table 6: Experimental results using CFB as label.

| Model | Length $P$ | MAE | RMSE | wMAPE |
|---|---|---|---|---|
| Informer | 7 days | 1.2566 | 4.2363 | 0.9480 |
| | 14 days | 0.8263 | 2.2853 | 0.8131 |
| | 30 days | 0.7799 | 1.9655 | 0.8007 |
| MQ-RNN | 7 days | 0.9053 | 3.0241 | 0.6829 |
| | 14 days | 0.7453 | 2.4314 | 0.7439 |
| | 30 days | 0.6913 | 2.1229 | 0.7189 |
| CRAFT | 7 days | 0.8480 | 2.8654 | 0.6706 |
| | 14 days | 0.7237 | 2.2696 | 0.7121 |
| | 30 days | 0.6895 | 2.0121 | 0.7078 |

In addition, we also explore the different uses of CFB based on Informer and MQ-RNN model. In the new experiment, we treat CFB features and label equally. Specifically, we define the time series forecasting problem as $\{\mathbf{Y}_P, \mathbf{C}_P\} = H(\mathbf{Y}_L, \mathbf{X}_t, \mathbf{C}_L)$, using CFB and label in the look-back window as input and CFB and label in prediction window as output. Experimental results are shown in Table 6. Compared with experimental results in Table 2, the effect of MQ-RNN is not significantly different, while the outcome of Informer deteriorates. Meanwhile, the wall-time training cost increased by 3-8 times.

# F Experimental Configuration

The time-series data is divided into training set, validation set, and testing set along the time dimension, which can avoid the problem of data traversal. The model is trained on the training set and hyperparameters are selected based on the descent of the loss function in the training set and the prediction error in the validation set. Based on this method, the search space and optimal parameters for each baseline model and CRAFT are as follows.

Table 7: The search space and optimal parameters for baseline models and CRAFT.

| Model | Parameter | Search space | Optimal value |
|---|---|---|---|
| MQRNN | encoder_hidden_layer | {1, 2, 4} | 2 |
| | encoder_hidden_size | {8, 16, 32, 64, 128, 256} | 128 |
| | decoder_hidden_size | {8, 16, 32, 64, 128, 256} | 128 |
| | learning_rate | {1e-4, 3e-4, 8e-4, 1e-3, 3e-3, 8e-3, 1e-2} | 3e-4 |
| Informer | heads_of_attention | {1, 2, 4, 8} | 8 |
| | hidden_size | {8, 16, 32, 64, 128, 256} | 64 |
| | inner_channel_size | {32, 64, 128, 256, 512, 1024, 2048} | 256 |
| | learning_rate | {1e-4, 3e-4, 8e-4, 1e-3, 3e-3, 8e-3, 1e-2} | 1e-3 |
| Autoformer | heads_of_attention | {1, 2, 4, 8} | 8 |
| | hidden_size | {8, 16, 32, 64, 128, 256} | 128 |
| | inner_channel_size | {32, 64, 128, 256, 512, 1024, 2048} | 512 |
| | learning_rate | {1e-4, 3e-4, 8e-4, 1e-3, 3e-3, 8e-3, 1e-2} | 8e-4 |
| Fedformer | heads_of_attention | {1, 2, 4, 8} | 8 |
| | hidden_size | {8, 16, 32, 64, 128, 256} | 128 |
| | inner_channel_size | {32, 64, 128, 256, 512, 1024, 2048} | 512 |
| | modes | {4, 8, 16, 32} | 16 |
| | learning_rate | {1e-4, 3e-4, 8e-4, 1e-3, 3e-3, 8e-3, 1e-2} | 1e-3 |
| TFT | heads_of_attention | {1, 2, 4, 8} | 2 |
| | hidden_size | {8, 16, 32, 64, 128, 256} | 128 |
| | learning_rate | {1e-4, 3e-4, 8e-4, 1e-3, 3e-3, 8e-3, 1e-2} | 3e-3 |
| DLinear | kernel_size | {3, 7, 15, 30} | {7, 7,15} |
| | individual | {True, False} | True |
| | learning_rate | {1e-4, 3e-4, 8e-4, 1e-3, 3e-3, 8e-3, 1e-2} | 3e-4 |
| Koopa | num_blocks | {1, 2, 4, 8} | 2 |
| | hidden_size | {8, 16, 32, 64, 128, 256, 512} | 128 |
| | dynamic_size | {4, 8, 16, 32, 64, 128, 256} | 64 |
| | seg_len | {3, 7, 15, 30} | {7, 15, 30} |
| | learning_rate | {1e-4, 3e-4, 8e-4, 1e-3, 3e-3, 8e-3, 1e-2} | 3e-4 |
| CRAFT | kernel_size | {3, 7, 15, 30} | {15, 15, 30} |
| | hidden_dim | {8, 16, 32, 64, 128, 256, 512} | 128 |
| | learning_rate | {1e-4, 3e-4, 8e-4, 1e-3, 3e-3, 8e-3, 1e-2} | 1e-3 |

{} represents the set of values. If there are three optimal values, they correspond to the case where the predicted lengths $P \in \{7, 14, 30\}$, respectively.