# OpenReview forum: "CRAFT: Time Series Forecasting with Cross-Future Behavior Awareness"
_ICLR.cc/2025/Conference — Submitted to ICLR 2025_

### Official Review · Reviewer_vwxz · 2024-11-03

**Soundness:** 2
**Presentation:** 2
**Contribution:** 2
**Rating:** 3
**Confidence:** 3

**Summary:**

The paper introduces CRAFT, a Cross-Future Behavior Awareness-based Time Series Forecasting model, designed to enhance time series predictions by leveraging future-aware information that occurs prior to the forecast period but has future relevance. CRAFT incorporates the trend of Cross-Future Behavior (CFB) into time series forecasting, addressing challenges like the sparsity and unpredictability of CFB through modules specifically crafted for key trend extraction, internal trend completion, and hierarchical guidance.Offline benchmarks and online A/B tests  validate CRAFT’s effectiveness, showing improvements over established models in both accuracy and practical forecasting metrics. The results highlight CRAFT’s potential to enhance forecasting performance in complex, real-world scenarios.

**Strengths:**

1. The paper presents an innovative approach to time series forecasting by incorporating Cross-Future Behavior (CFB), which effectively captures future events that impact the forecast. This concept is novel and valuable, especially for applications like e-commerce and hotel bookings where advance information is critical.
2. The proposed CRAFT framework is comprehensive, utilizing three modules—Koopman Predictor Module (KPM), Internal Trend Mining Module (ITM), and External Trend Guide Module (ETG)—to tackle challenges related to CFB, such as sparsity and trend ambiguity, thereby improving forecasting accuracy.
3. The inclusion of demand-constrained loss is a notable strength, as it aligns the model’s predictions more closely with practical constraints in real-time applications.

**Weaknesses:**

1. Some citations are inaccurate. For example, in line 54, the series decomposition method used in DLinear originates from Autoformer. Additionally, DLinear decomposes the time series into trend and seasonal components, not trend and remainder components.
2. Some parts of the writing lack clarity. For instance, in Table 2, the description “Comparative forecasting results with the look-back window length $L$ and prediction window length $P$ respectively” is unclear because the look-back window length $L$ is not displayed in the table.
3. The paper contains inconsistent notation. For example, near line 152, $C_t = \{C_{t−L+1:t}, C_{t+1:t+P}\}$ implies an infinite inclusion by reusing the same $C$. Additionally, forecasting length is denoted by both $K$ and $L$ in Section 5, leading to confusion.

**Questions:**

1. TSMixer also uses future features for time series forecasting and evaluated the model on the M5 dataset, where auxiliary information like promotions and vacations are provided, similar to the “CFB” in your work. How does your method compare to TSMixer in this context?
2. The introduction refers to two main challenges, but these are not clearly defined. Could you elaborate on these challenges?
3. In Table 2, why does a longer prediction window generally result in lower error? Typically, a larger forecasting horizon is more challenging due to increased uncertainty.

---

> ### Author Response · Authors · 2024-11-28
> **Response to Reviewer vwxz (Part 1/n)**
>
> > Summary
>
> We thank the reviewer for reviewing our manuscript and giving a lot of significiant suggestions. We have responded to the reviewer's comments below and have addressed them accordingly in the uploaded revision of the paper. **Given the improvement suggestion by all reviewers, we have re-modified our manuscript and uploaded the revised version. We look forward to the reviewer's reassessment of our manuscript.**
>
> > **Response to Weakness 1**: about the citation problem
>
> Thanks for your careful suggestion. We have checked the original DLinear paper: "DLinear is a combination of a Decomposition scheme used in Autoformer and FEDformer with linear layers. It first decomposes a raw data input into a trend component by a moving average kernel and a remainder (seasonal) component." **In the original paper of DLinear, the auhtors do use the expression of "remainder component".**
>
> > **Response to Weakness 2 and Weakness 3**: some parts of the writing lack clarity and the inconsistent notations
>
> Thank you for pointing out the writing and expression problems in this article. We have rechecked the article and sorted out the confusing content and corresponding corrections as follows :
>
> 1) *Original text* : Table 2: Comparative forecasting results with the look-back window length of L $\in$ {30, 90, 180} and prediction window length of P $\in$ {7, 14, 30} respectively. The best results are highlighted in bold and the second best results are highlighted with a underline.
>
>    *Reason for correction*: lack clarity, the look-back window length L is not displayed in the table.
>    *Corrected text*: Table 2: Comparative forecasting results with the ~~look-back window length of L $\in$ {30, 90, 180} and~~  prediction window length of P ∈ {7, 14, 30} respectively, correspond one-to-one with the look back window L ∈ {30, 90, 180}. The unit of length is days. The best results are highlighted in bold and the second best results are highlighted with a underline.
>
>    In addition, column name “Length” in Table 2 will be changed to “Length P”
>
> 2) *Original text* :
>
> a) etc. $C_t=\{C_{t-L+1:t},C_{t+1:t+p}\}$, where $C_{t-L+1:t}$ is CFB in the look-back window and $C_{t+1:t+p}$ is CFB in the prediction window. (Near line 152)
>
> b) Time series forecasting (TSF) with Cross-Future Behavior (CFB) can be defined as $Y_{t+1:t+p}=H(Y_{t-L+1:t}, X_{t}, C_{t})$. (Near line 148)
>
> c) $X_t$ is covariate features, $C_t$ is the CFB feature and H is the prediction function to be learned. (Near line 150)
>
> d) KPM module aims to transfer the future trend information from CFB feature $C_{t}=\{C_{L}, C_{P}\}$ to labels $Y_{P}$ . (Near line 200)
>
> *Reason for correction* : reuse the same symbol $C_t$ in set symbol and elements within a set.
>
> *Corrected text* :
>
> a) etc. $C_{𝕋}=\{C_{t-L+1:t},C_{t+1:t+p}\}$, where $C_{t-L+1:t}$ is CFB in the look-back window and $C_{t+1:t+p}$ is CFB in the prediction window.
>
> b) Time series forecasting (TSF) with Cross-Future Behavior (CFB) can be defined as $Y_{t+1:t+p}=H(Y_{t-L+1:t}, X_{𝕋}, C_{𝕋})$.
>
> c) $X_𝕋$ is covariate features, $C_𝕋$ is the CFB feature and H is the prediction function to be learned.
>
> d) The same applies to the value of $C_t$ in the following text. KPM module aims to transfer the future trend information from CFB feature $C_{𝕋}=\{C_{L}, C_{P}\}$ to labels $Y_{P}$ .

---

> ### Author Response · Authors · 2024-11-28
> **Response to Reviewer vwxz (Part 2/n)**
>
> 3) *Original text* :
>
> a)   which is designed as a linear network:  (Near line 244)
>
> $$
> {ZC}_P^T=Complete\\_network(ZC^T)  \ \ \ \ \ \ \ \ \ \ \ \ \  Eq(5)
> $$
>
> b)  Finally, the latent vector $\hat{ZC}_P^T$ and $\hat{ZY}_P^T$ are converted into target predicted values with decoder: $\hat{C}_P^T=Decoder(\hat{ZC}_P^T),\hat{Y}_P^T=Decoder(\hat{ZY}_P^T)$. (Near line 256)
>
>
> c)  The final result is calculated by adding these two values with the reminder predicted values (acquired with the linear mapping of $Y_L^R$, $C_L^R$ in Figure 3): (Near line 258)
>
> $$
> \hat{C}_P=\hat{C}_P^T + \hat{C}_P^R,  \hat{Y}_P=\hat{Y}_P^T + \hat{Y}_P^R \ \ \ \ \ \ \ \ \ \ \ \ \  Eq(7)
> $$
>
> d) The intact matrix expression of prediction window of cross-future behavior $\hat{ZC}_P$ (Near line 340)
>
> e)  which corresponding to the recovery loss of $\hat{C}_{KP}$ in the KPM module, the prediction error of $\hat{C}_P$ in the ITM module, the reconstruction drift of $\hat{Y}_P$ at the ETG module and the forecast deviation of label $\hat{Y}_P$ respectively. (Near line 347)
>
> *Reason for correction*:  Confused $\hat{ZC}\_P^T$ and $\hat{ZC}\_{TP}^T$, $\hat{C}\_P$ and $\hat{C}\_{TP}$
>
> *Corrected text*:
>
> a) which is designed as a linear network:  (Near line 244)
>     $$\hat{ZC}\_{TP}^T=Complete\\_network(ZC^T)  \ \ \ \ \ \ \ \ \ \ \ \ \  Eq(5)$$  b)  Finally, the latent vector $\hat{ZC}\_{TP}^T$ and $\hat{ZY}\_P^T$ are converted into target predicted values with decoder:  $\hat{C}\_{TP}^T=Decoder(\hat{ZC}\_{TP}^T),\hat{Y}\_P^T=Decoder(\hat{ZY}\_P^T).$
>
> c)  The final result is calculated by adding these two values with the reminder predicted values (acquired with the linear mapping of $Y\_L^R$, $C\_L^R$ in Figure 3): (Near line 258)
> $$\hat{C}\_{TP}=\hat{C}\_{TP}^T + \hat{C}\_{TP}^R,  \hat{Y}\_P=\hat{Y}\_P^T + \hat{Y}\_P^R \ \ \ \ \ \ \ \ \ \ \ \ \  Eq(7)$$  d) The intact matrix expression of prediction window of cross-future behavior $\hat{C}\_{TP}$. (Near line 340)
>
> e)  which corresponding to the recovery loss of  $\hat{C}\_{P}$ in the KPM module, the prediction error of  $\hat{C}\_{TP}$ in the ITM module, the reconstruction drift of $\hat{Y}\_P$ at the ETG module and the forecast deviation of label $\hat{Y}\_P$ respectively. (Near line 347)
>
> Synchronously, the symbols in the figures of the paper have also been modified accordingly.

---

> > ### Author Response · Authors · 2024-11-28
> > **Response to Reviewer vwxz (Part 3/n)**
> >
> > > **Response to Question 1**: can our method compare with TSMixer?
> >
> > Thank you for providing the idea for model comparison, we have also considered comparing on public datasets. However, **the existing public datasets do not have the features of CFB, thus our model cannot be validated in public datasets**, which is regrettable but cannot do anything to help now. The auxiliary information in the M5 dataset, such as promotions and vacations, belong to label features that are fully known in the future, which is different from partial known numerical CFB. What’s more, although it cannot be validated on public datasets, **we still have these auxiliary information in our own dataset, such as holidays, festivals and major exams**. These features serve as covariate features X, which are applied in MQRNN, TFT, and not very compatible with other models.
> >
> > > **Response to Question 2**: about the definition to two main challenges
> >
> > Thanks for your question. Limited by the pages, we summary two challenges as **1) CFB is sparse and partial**. Consumers can book items at any time, causing CFB to remain fully observed until the last minute. Thus, if CFB is simply incorporated into the model, the prediction model may be unable to apply CFB features correctly and even make incorrect predictions due to CFB. And **2) The trend of CFB is unobvious**. As shown in Figure 2, compared with the sales trend in a business district, the sales trend in an individual hotel is unobvious. CFB has the same nature, and the trend in an individual hotel is much unobvious compared with that in a high-level business district. We have supplemented the definition to two main challenges in the appendix A with blue font:
> >
> > "However, the use of CFB in TSF problem faces two main challenges:1) **CFB is sparse and partial**. Consumers can book items at any time, causing CFB to remain fully observed until the last minute. As shown in Figure A(d), half of the data in the cross-future matrix cannot be obtained. When making predictions, the data for row T-0 can only be fully obtained at the end of the day, meaning only rows T-1 to T-n are available for forecasting, showing that CFB is sparser and smaller than other features such as hotel historical sales. Thus, if CFB is simply incorporated into the model, the prediction model may be unable to apply CFB features correctly and even make incorrect predictions due to CFB. 2) **The trend of CFB is unobvious**. As shown in Figure 2, compared with the sales trend in a high-level business district, the sales trend in an individual hotel is unobvious. CFB has the same nature, and the trend in an individual hotel is much unobvious compared with that in a high-level business district. CFB can reflect future trends to a certain extent, however, the coverage of CFB for some hotels may be too atypical to reveal a trend. Sales at higher-level district are much representative than in individual hotels. If there is a rising trend at a higher level, it is also possible that the sales of individual hotels may increase. How to leverage the sales trends of the higher level to guide the trends exploration for individual hotels is the second challenge."
> >
> > > **Response to Question3**: about the predition results in Table 2
> >
> > Thank you for reminding, this may indeed confuse readers. In general, the longer the prediction window, the higher the prediction uncertainty and the prediction error. **The reason why our model yielded seemingly contradictory conclusions is the composition of our prediction window.** Regardless of the length of the prediction window, holiday data is included. Due to the differences between holiday patterns and daily pattern, predicting during holidays is more difficult. As the length of the prediction window increases, the proportion of holiday data decreases, resulting in a phenomenon where the longer the prediction window length, the smaller the prediction error. We described it in section 5.1.1  Dataset : *To reflect the model effects on different data distributions objectively, the prediction window we cover to verify the model’s effectiveness contains both holidays and daily events.*
> >
> > To avoid the same confusion, we will include the above analysis results in Section 5.2.1 Comparison With Baselines (Near line 470) :  *In the experimental results, the longer the prediction window, the smaller the prediction error. The reason is that the prediction windows consist of daily data and holiday data, and holiday patterns are more difficult to predict. As the length of the prediction window increases, the proportion of holiday data decreases and the overall prediction error reduces.*

---

> > > ### Author Response · Authors · 2024-12-02
> > > **Response to Reviewer vwxz (Part 4/n)**
> > >
> > > Dear Reviewer vwxz,
> > >
> > > Thank you again for your time and effort in reviewing our paper.
> > >
> > > As the new Discussion Period ends on December 2nd at 24:00 AoE, we kindly remind you to review our revised manuscript and responses. We are eagerly awaiting your feedback.

---

> > > > ### Author Response · Authors · 2024-12-03
> > > > **Response to Reviewer vwxz (Part 5/n)**
> > > >
> > > > Dear Reviewer vwxz,
> > > >
> > > > Due to the discussion between the author and the reviewer being near its conclusion, we would like to know whether our response has addressed your main concerns. If so, we kindly ask for your reconsideration of the score. Should you have any further advice on the paper and/or our rebuttal, please let us know and we will be more than happy to engage in more discussion and paper improvements.
> > > >
> > > > Thank you so much for devoting time to improving our work!

---

> > > > > ### Author Response · Authors · 2024-12-03
> > > > > **Response to Reviewer vwxz (Part 6/n)**
> > > > >
> > > > > Dear reviewer vwxz,
> > > > > As the discussion time is coming to an end, we eagerly await your reply. Thanks for your time.

---

### Official Review · Reviewer_U8vN · 2024-11-03

**Soundness:** 2
**Presentation:** 2
**Contribution:** 2
**Rating:** 3
**Confidence:** 5

**Summary:**

This paper focuses on the application of time series in the field of e-commerce, attempting to incorporate e-commerce characteristics into the model building process and proposing the concepts of CFB and the CRAFT model. In the model design process, the authors drew inspiration from a lot of previous work, combining them to form the CRAFT model. Finally, some experiments were conducted by the authors to demonstrate the effectiveness of the proposed model.

**Strengths:**

1. The field of time series forecasting that this paper focuses on is worth studying.

2. The structure of the article is relatively complete.

**Weaknesses:**

1. The research motivation has significant issues. Although there are instances in e-commerce where actions are taken at a past moment for a future one, it is evident that they are entirely corresponding. Changes in final labels can be understood as minor variations based on the booking situation. Therefore, even though the actual future events have not occurred at the current time, crucial actions that influence the future have already taken place in history. Introducing the so-called CFB data essentially involves bringing in a part of real future data, thus constituting data leakage. This approach is fundamentally distinct from previous attempts to introduce more features because earlier works aimed at extracting or learning more features from historical time series data without any data leakage.

2. There are serious shortcomings in the survey of time series forecasting methods. The author noted PatchTST from ICLR 2023 but overlooked contemporaneous models such as TimesNet (CNN-Based)[1] and MICN (CNN-Based)[2]. The author mentioned Koopa from NeurIPS 2023 but failed to acknowledge concurrent models like WITRAN (RNN-Based)[3] and Basisformer (Attention-Based)[4]. Furthermore, no attention was given to any time series forecasting methods from ICLR 2024, such as FITS (MLP-Based)[5], TimeMixer (MLP-Based)[6], ModernTCN (CNN-Based)[7], and iTransformer (Attention-Based)[8].

3. The model lacks innovation, as the designs in KPM, ITM, and ETG are primarily derived from previous works, making it challenging to identify entirely independently innovative content.

4. Code was not provided, leading to poor reproducibility of experimental results.

5. The experiments are insufficient as the methods compared are relatively older works and do not comprehensively prove the efficacy of the experiments. Additionally, the method proposed by the author can only be applied under the assumption of data leakage through CFB, making its applicability weak for scenarios with only labels. Moreover, in Table 2, it is evident that in certain instances, utilizing only labels yields better results compared to using CFB in nearly every baseline method. This further indicates that the effectiveness of CFB is not necessarily proven.

6. The author did not present the full search space for reproducing the baselines. Results can vary significantly for the same parameters on different platforms. Therefore, a fair approach would involve setting a consistent search space for all methods on the same platform and determining the best parameters for each model on various tasks using a validation set. I noticed that the experimental platform used by the author is inconsistent with the platforms of the compared methods, so the author should address this to demonstrate the credibility of the experiments. Otherwise, it is challenging to eliminate the significant impact of parameter selection on the experimental conclusions.

References:

[1] Wu, H., Hu, T., Liu, Y., Zhou, H., Wang, J., & Long, M. (2023). TimesNet: Temporal 2D-Variation Modeling for General Time Series Analysis. In The Eleventh International Conference on Learning Representations.

[2] Wang, H., Peng, J., Huang, F., Wang, J., Chen, J., & Xiao, Y. (2023). Micn: Multi-scale local and global context modeling for long-term series forecasting. In The Eleventh International Conference on Learning Representations.

[3] Jia, Y., Lin, Y., Hao, X., Lin, Y., Guo, S., & Wan, H. (2023). WITRAN: Water-wave Information Transmission and Recurrent Acceleration Network for Long-range Time Series Forecasting. In Thirty-seventh Conference on Neural Information Processing Systems.

[4] Ni, Z., Yu, H., Liu, S., Li, J., & Lin, W. (2023). BasisFormer: Attention-based Time Series Forecasting with Learnable and Interpretable Basis. In Thirty-seventh Conference on Neural Information Processing Systems.

[5] Xu, Z., Zeng, A., & Xu, Q. (2024). FITS: Modeling Time Series with $10 k $ Parameters. In The Twelfth International Conference on Learning Representations.

[6] Wang, S., Wu, H., Shi, X., Hu, T., Luo, H., Ma, L., ... & ZHOU, J. (2024). TimeMixer: Decomposable Multiscale Mixing for Time Series Forecasting. In The Twelfth International Conference on Learning Representations.

[7] Luo, D., & Wang, X. (2024). Moderntcn: A modern pure convolution structure for general time series analysis. In The Twelfth International Conference on Learning Representations.

[8] Liu, Y., Hu, T., Zhang, H., Wu, H., Wang, S., Ma, L., & Long, M. (2024). iTransformer: Inverted Transformers Are Effective for Time Series Forecasting. In The Twelfth International Conference on Learning Representations.

**Questions:**

1. Can the author provide the search space and optimal parameters for all methods in all tasks?

2. Can the author provide the code and a small portion of the dataset for reproducibility purposes?

---

> ### Author Response · Authors · 2024-11-28
> **Response to Reviewer U8vN (Part 1/n)**
>
> > Summary
>
> We thank reviewer U8vN for reviewing our manuscript and pointing out a set of significiant inquiries. We have responded to the reviewer's comments below and have addressed them accordingly in the uploaded revision of the paper. **Given the improvement suggestion by all reviewers, we have re-modified our manuscript and uploaded the revised version. We look forward to the reviewer's reassessment of our manuscript.**
>
> > **Response to Weakness 1**: about the data leakage problem
>
> Thanks for your concerns. However, we have to declare that **CFB are a normal prior features rather than labels**.  Features in machine learning refers to individual measurable properties or characteristics of a phenomenon being observed, using as input variables for model prediction. Typically, features should satisfy the measurable, independent, and varied types condition. Labels are the outputs that the models are trained to predict, and **the labels should depend on the input features. The depend property between features and labeles is the theoretical foundation for reasonable machine learning model construction.**
>
> On the other hand, in machine learning, there is a important research area, named **feature engineering. The aim of feature engineering is to discovery significiant features to improve the performance of machine learning model.** Our research to discovery CFB feature belongs to this area.
>
> Returning to the specific field of time series forecasting, many existing works are also attempting to introduce more features to improve the performance of time series forecasting. Wen et al. [1] classified covariate features in TSF into dynamic historical, known future, and static variables. Chen et al. [2] used auxiliary information like promotions and vacations in M5 dataset for prediction. **These additional featues are also similar to CFB features, and they do not involve data leakage problem.**
>
> Thus, **we can not agree with your doubts about data leakage. CFB features we introduce in our paper are prior features that we can acquire and use before the current prediction timepoint.** Table 2 predicts the labels before 7, 14 and 30 days at most. CFB features is related to the forecasting results. The relathionship between CFB features and forecasting results is the theoretical foundation for our research.
>
> [1] Wen R, Torkkola K, Narayanaswamy B, et al. A multi-horizon quantile recurrent forecaster[J]. arXiv preprint arXiv:1711.11053, 2017.
> [2] Chen S A, Li C L, Yoder N, et al. Tsmixer: An all-mlp architecture for time series forecasting[J]. arXiv preprint arXiv:2303.06053, 2023.

---

> ### Author Response · Authors · 2024-11-28
> **Response to Reviewer U8vN (Part 2/n)**
>
> > **Response to Weakness 2**: about the citation
>
> Thanks for your suggestions. According to the summary in first paragraph in the Introdcution Section and the summary in the Related Work. Backbone for time series forecasting Section, you can know that we also obey the consistent idea with you about the classification of existing time series forecasting method, i.e., the CNNs, RNNs, and Transformer-based methods.
>
> Recently, more and more research focues on time series forecasting problem. We use "time series forecasting" as the keyword to search in Web of Science and DBLP. According to our statistics, the numbers of targeted paper in Web of Science are 4252 (2024), 4839 (2023), 4824 (2022), 4279 (2021), 3692 (2020), and 3220 (2019). The numbers of targeted paper in DBLP are 2024 (688), 2023 (586), 2022 (436), 2021 (366), 2020 (300), and 2019 (196). The numbers of targeted paper published in machine learning related conference are 27 (ICLR), 30 (AAAI), 20 (IJCAI), 20 (ICML), and 32 (NeurIPS).
>
> In any paper, **there is impossible to cite all related papers totally. What we can do is to obey the correct classification criteria and list the latest and most relevant works. In addition, the recommended eight papers, especially the [3-4] paper, do not have strong authority and relevance.**
>
> More importanly, **the reviewer points out that "no attention was given to any time series
> forecasting methods from ICLR 2024", but we have cited five time series
> forecasting methods methods from ICLR 2024**:
>
> [1] Rasul K, Bennett A, Vicente P, et al. VQ-TR: Vector Quantized Attention for Time Series Forecasting[C]//The Twelfth International Conference on Learning Representations. 2024.
>
> [2] Zhao L, Shen Y. Rethinking Channel Dependence for Multivariate Time Series Forecasting: Learning from Leading Indicators[C]//The Twelfth International Conference on Learning Representations. 2024.
>
> [3] Li Y, Chen W, Hu X, et al. Transformer-Modulated Diffusion Models for Probabilistic Multivariate Time Series Forecasting[C]//The Twelfth International Conference on Learning Representations. 2024.
>
> [4] Wang X, Zhou T, Wen Q, et al. CARD: Channel aligned robust blend transformer for time series forecasting[C]//The Twelfth International Conference on Learning Representations. 2024.
>
> [5] Nie Y, Nguyen N H, Sinthong P, et al. A time series is worth 64 words: Long-term forecasting with transformers[C]//The Twelfth International Conference on Learning Representations. 2024.

---

> > ### Author Response · Authors · 2024-11-28
> > **Response to Reviewer U8vN (Part 3/n)**
> >
> > > **Response to Weakness 3**: about the innovation
> >
> > First of all, **the core innovation of our article is to define the cross-future behavior features and utilize the trend of cross-future behavior to mine the trend of time series data to be predicted. Secondly, our model is also innovative.**
> > **All research work is developed based on previous works**, we don’t know how to define derive here. All Transformer-based models are based on attention structures, and CNN, DNN, and RNN also serve as the foundation for many models. For instance, DLinear only employs two one-layer linear networks, but it outperforms complex Transformer-based models that existed at the time in most cases by a large margin. **Returning to our model CRAFT, the KPM and ITM modules are distinct from existing works.**
> >
> > 1) For the KPM module, although it is inspired by the Koopa model, the module structure also has significant differences from the Koopa model. From a target perspective, KPM module aims to transfer the future trend information from CFB feature $C_{𝒯}=\{C_L, C_P\}$ to labels, while Koopa model infers future information based on look-back window. From the perspective of structural design, the Koopa model has two main modules : Time-invariant KP and Time-variant KP. The Koopman operator $K_{inv}$ in Time-invariant KP module is a learnable parameter, which regards the embedding of lookback and forecast window $Z_{back}$, $Z_{fore}$ as running snapshot pairs. That is $Z_{fore} = K_{inv} Z_{back}$, where $Z_{fore}$ is unknown during inference, and $K_{inv}$ is learned based on historical data. Time-variant KP module utilizes localized snapshots in the look-back window. Namely,
> >
> > $$
> > Z_{back\_b}=[Z_1, Z_2, ..., Z_{\frac{T}{S}-1}],  Z_{back\_f}=[Z_2, Z_3, ..., Z_{\frac{T}{S}}], \\ K_{var}=Z_{back\_b} Z_{back\_f}^T$$ However, the Koopman matrix $K_C$ in KPM module is calculated by $K_C=Z_{back}Z_{fore}^T$, Where $Z_{fore}$ is completely known. In addition, to ensure that $K_C$ is meaningful, we construct a recovery loss to constrain the decoder to restore the original data based on the embedding output by the encoder.
> > <small>*Ps: for ease of comparison, look-back and prediction window are unified referred as $Z_{back}$ and $Z_{fore}$, which differs from the symbols used in the text.*</small>
> > 2) The ITM module is completely self-designed, based on the need to complete linear mapping between the look-back window and prediction window. If possible, please provide relevant previous work.
> >
> > > **Response to Weakness 4**: about the code
> >
> > Thanks for your remind. We have declared in the abstract parts that **"Our dataset and code will be released after formal publication."**

---

> > > ### Author Response · Authors · 2024-11-28
> > > **Response to Reviewer U8vN (Part 4/n)**
> > >
> > > > **Response to Weakness 5**: about the sufficiency of the experiment : the comparative methods are relatively older works; assumption of data leakage through CFB; the effect of baseline with CFB
> > >
> > > We will response your series inquiries one by one:
> > >
> > > **The comparative methods are relatively older works**: We compare our proposed CRAFT method with seven baseline methods, including MQ-RNN (Wen et al., 2017), Informer (Zhou et al., 2021), DLinear (Zeng et al., 2023), Koopa (Liu et al., 2024), TFT (Lim et al., 2021), Autoformer (Wu et al., 2021) and Fedformer (Zhou et al., 2022).
> > > 1. **These seven models represent different types of temporal models**, MQRNN is based on RNN, DLinear is based on linear layers, Koopa is based on Koopman theory, Informer, Autoformer, Fedformer, and TFT are based on Transformer.
> > > 2. **These models include recognized models and sota models.** Koopa model is the sota model of 2024, and other models are also widely recognized as baseline models for comparison. In ICLR 2024, Cheng et al. [1] used Informer, Autoformer, and Fedformer to evaluate RobustTSF on the power dataset. The comparative model used in Liu et al. [2] also includes DLinear, Fedformer, Autoformer, Informer, and three other models: Reformer (Kitaev et al. (2020)), Pyraformer (Liu et al. (2021)) and LogTrans (Li et al. (2019)), the comparative models adopted in Wu et al. [3] are DLinear, Fedformer, Informer, Autoformer, and Crossformer (Zhang and Yan, 2022)).
> > > 3. **The year of the model is not a mandatory option.** In ICLR 2024, Xu et al. [4] utilized TS2Vec (Yue et al., 2022), TNC (Tonekaboni et al., 2021), CPC (Oord et al., 2018) and SimCLR (Chen et al., 2020) as benchmark models for comparison, the benchmark models introduced in Sun et al. [5] are MC-dropout (Gal&Ghahramani (2016a)), BJRNN (Alaa&VanDer Schaar (2020)), and CF-RNN (Stankeviciute et al. (2021))
> > >
> > > **Assumption of data leakage through CFB**:
> > >  Firstly, **CFB does not have the issue of data traversal,** which has been explained in Weakness 1. Secondly, we want to declare that **our proposed CRAFT method is designed for the effectiveness use of CFB features**. Every machine learning method is designed to fit different scenarios.
> > >
> > > **The effect of baseline with CFB**:
> > > As for the experimental results in Table 2, for the baseline methods, compared with the scenario with only lables, the scenario with CFB features does not acquire significant advantages in performance. We have discussed this phenomena in Section 5.2.1 (Comparison with Baselines): "Compared to the original model, directly integrating CFB into the existing framework does not yield significant performance enhancements. In some cases, it even leads to performance degradation. The experimental results confirm that, despite its indispensable role, effectively applying CFB presents considerable challenges." **This phenomena also provides the motivation for the design of our CRAFT method.**
> > >
> > > [1] Cheng H, Wen Q, Liu Y, et al. RobustTSF: Towards Theory and Design of Robust Time Series Forecasting with Anomalies[C]//The Twelfth International Conference on Learning Representations.2024.
> > >
> > > [2] Liu X, Chen D, Wei W, et al. Interpretable Sparse System Identification: Beyond Recent Deep Learning Techniques on Time-Series Prediction[C]//The Twelfth International Conference on Learning Representations.2024.
> > >
> > > [3] Wu D, Hu J Y C, Li W, et al. STanHop: Sparse Tandem Hopfield Model for Memory-Enhanced Time Series Prediction[C]//The Twelfth International Conference on Learning Representations.2024.
> > >
> > > [4] Xu M, Moreno A, Wei H, et al. REBAR: Retrieval-Based Reconstruction for Time-series Contrastive Learning[C]//The Twelfth International Conference on Learning Representations.2024.
> > >
> > > [5] Sun S H, Yu R. Copula Conformal prediction for multi-step time series prediction[C]//The Twelfth International Conference on Learning Representations. 2023.
> > >
> > > > **Response to Weakness 6**: about the comparative environment of baseline methods
> > >
> > > Thanks for your suggestion. We are sorry that we neglect the search space and optimal parameters for baseline methods. **We have supplemented these in our upload pdf in Appendix F Section (Experimental Configuration) with blue color.**
> > >
> > > As for the experimental platform for our proposed CRAFT method and the baseline methods, they do all conduct experiment on the same platform. As shown in Section 5.1.3 (Implementation): **"All experiments are implemented with Python 3.8.5 and Pytorch 1.12.1 We conduct them on the cloud servers with two NVIDIA Tesla T4 GPUs with 16GB VRAM each."** We are sorry for your misunderstanding.

---

> > > > ### Author Response · Authors · 2024-11-28
> > > > **Response to Reviewer U8vN (Part 5/n)**
> > > >
> > > > > **Response to Question 1**: provide the search space and optimal parameters for all methods in all tasks
> > > >
> > > > The search space and optimal parameters for baseline models and craft are as follows :
> > > > | Model       | Parameter            | Search space                               | Optimal value |
> > > > |-------------|----------------------|--------------------------------------------|---------------|
> > > > | MQRNN       | encoder_hidden_layer | {1, 2, 4}                                  | 2             |
> > > > |             | encoder_hidden_size  | {8, 16, 32, 64, 128, 256}                  | 128           |
> > > > |             | decoder_hidden_size  | {8, 16, 32, 64, 128, 256}                  | 128           |
> > > > |             | learning_rate        | {1e-4, 3e-4, 8e-4, 1e-3, 3e-3, 8e-3, 1e-2} | 3e-4          |
> > > > | Informer    | heads_of_attention   | {1, 2, 4, 8}                               | 8             |
> > > > |             | hidden_size          | {8, 16, 32, 64, 128, 256, 512}             | 64            |
> > > > |             | inner_channel_size   | {32, 64, 128, 256, 512, 1024, 2048}        | 256           |
> > > > |             | learning_rate        | {1e-4, 3e-4, 8e-4, 1e-3, 3e-3, 8e-3, 1e-2} | 1e-3          |
> > > > | Autoformer  | heads_of_attention   | {1, 2, 4, 8}                               | 8             |
> > > > |             | hidden_size          | {8, 16, 32, 64, 128, 256, 512}             | 128           |
> > > > |             | inner_channel_size   | {32, 64, 128, 256, 512, 1024, 2048}        | 512           |
> > > > |             | learning_rate        | {1e-4, 3e-4, 8e-4, 1e-3, 3e-3, 8e-3, 1e-2} | 8e-4          |
> > > > | Fedformer   | heads_of_attention   | {1, 2, 4, 8}                               | 8             |
> > > > |             | hidden_size          | {8, 16, 32, 64, 128, 256, 512}             | 128           |
> > > > |             | inner_channel_size   | {32, 64, 128, 256, 512, 1024, 2048}        | 512           |
> > > > |             | modes                | {4, 8, 16, 32}                             | 16            |
> > > > |             | learning_rate        | {1e-4, 3e-4, 8e-4, 1e-3, 3e-3, 8e-3, 1e-2} | 1e-3          |
> > > > | TFT         | heads_of_attention   | {1, 2, 4}                                  | 2             |
> > > > |             | hidden_size          | {8, 16, 32, 64, 128, 256, 512}             | 128           |
> > > > |             | learning_rate        | {1e-4, 3e-4, 8e-4, 1e-3, 3e-3, 8e-3, 1e-2} | 3e-3          |
> > > > | DLinear     | kernel_size          | {3,7,15,30}                                | {7,7,15}      |
> > > > |             | individual           | {True, False}                              | True          |
> > > > |             | learning_rate        | {1e-4, 3e-4, 8e-4, 1e-3, 3e-3, 8e-3, 1e-2} | 3e-4          |
> > > > | Koopa       | num_blocks           | {1, 2, 4, 8}                               | 2             |
> > > > |             | hidden_dim           | {8, 16, 32, 64, 128, 256, 512}             | 128           |
> > > > |             | dynamic_dim          | {4, 8, 16, 32, 64, 128, 256}               | 64            |
> > > > |             | seg_len              | {3, 7, 15, 30}                             | {7, 15, 30}   |
> > > > |             | learning_rate        | {1e-4, 3e-4, 8e-4, 1e-3, 3e-3, 8e-3, 1e-2} | 3e-4          |
> > > > | CRAFT       | kernel_size          | {3,7,15,30}                                | {15, 15, 30}  |
> > > > |             | hidden_dim           | {8, 16, 32, 64, 128, 256, 512}             | 128           |
> > > > |             | learning_rate        | {1e-4, 3e-4, 8e-4, 1e-3, 3e-3, 8e-3, 1e-2} | 1e-3          |
> > > > |             |                      |                                            |               |
> > > >
> > > >
> > > > *{} represents the set of values. If there are three optimal values, they correspond to the case where the predicted lengths P $\in$ {7, 14, 30}, respectively.*
> > > > In addition to the internal hyperparameters of the model mentioned above, we also provided the configuration of the model in section 5.1.3 IMPLEMENTATION of the paper, and offered a relevant analysis of the loss coefficient and the parameter in solving koopman matrix that has a significant impact on CRAFT in section E.2 HYPER PARAMETER ANALYSIS. Besides, we apologize for not clearly stating the optimal choice of kernel size in section 5.1.3, we will remove it from the text. And we will update the paramter configuration in the appendix F EXPERIMENTAL CONFIGURATION.
> > > >
> > > > > **Response to Question 2**: provide the code and a small portion of the dataset
> > > >
> > > > Thanks for your remind. We have declared in the abstract parts that **"Our dataset and code will be released after formal publication."**

---

> > > > > ### Author Response · Authors · 2024-12-02
> > > > > **Response to Reviewer U8vN (Part 6/n)**
> > > > >
> > > > > Dear Reviewer U8vN,
> > > > >
> > > > > Thank you again for your time and effort in reviewing our paper.
> > > > >
> > > > > As the new Discussion Period ends on December 2nd at 24:00 AoE, we kindly remind you to review our revised manuscript and responses. We are eagerly awaiting your feedback.

---

> > > > > > ### Comment · Reviewer_U8vN · 2024-12-03
> > > > > >
> > > > > > Thank you for the response. My concerns have not been fully addressed. In terms of innovation, the authors acknowledge that 'the KPM and ITM modules are distinct from existing works' which indicate a lack of novelty in this paper. Regarding experimentation, the methods compared by the authors are inadequate. The methods I outlined in the original review are among the most recent representative state-of-the-art (SOTA) methods, yet I have not seen a comparative analysis of the authors' results, making it difficult to support the conclusions of this paper.
> > > > > >
> > > > > > Concerning parameter search, although the authors have made some efforts, there are still shortcomings. For instance, the search range for 'e_layer' and 'd_layer' in Informer has not been specified. As I emphasized in the original review: 'Results can vary significantly for the same parameters on different platforms. Therefore, a fair approach would involve setting a consistent search space for all methods on the same platform and determining the best parameters for each model on various tasks using a validation set.' The authors only conducted partial parameter searches related to the model and overlooked other crucial parameters. Consequently, the experimental results based on these factors are not convincing.
> > > > > >
> > > > > > Taking into account the innovation, quality, and clarity of this paper, as well as the authors addressing few of my concerns, I will reconsider the score.

---

> > > > > > > ### Author Response · Authors · 2024-12-03
> > > > > > > **Response to Reviewer U8vN (Part 7/n)**
> > > > > > >
> > > > > > > Dear  U8vN,
> > > > > > >
> > > > > > > Thanks for your timely responses.  Here are the responses to each point:
> > > > > > >
> > > > > > > ● **Innovation concerning**
> > > > > > >
> > > > > > > As mentioned in your response, 'the authors acknowledge that' the KPM and ITM modules are distinct from existing works', **which precisely demonstrates the innovation of our paper: we define the CFB feature innovatively and apply the CFB feature to time series forecasting for the first time.** And a large number of experiments have shown that directly integrating CFB into the existing framework does not yield significant performance enhancements, and craft can effectively apply CFB features.
> > > > > > >
> > > > > > > ● **Regarding experimentation**
> > > > > > >
> > > > > > > 1）DLinear (Zeng et al., 2023) and Koopa (Liu et al., 2024) used in our paper are typical representatives of the sota model in the past two years. **The Koopa (Liu et al., 2024) paper also used the papers TimesNet and MICN provided by the reviewer, but they were not better than Koopa.** In addition, the [3-4] paper, do not have strong authority and relevance.
> > > > > > >
> > > > > > > 2）The official email on November 26 clearly stated that "Reviewers are instructed to not ask for significant experiments, and area chairs are instructed to discard significant reviewer requests." As a Reviewer you could have explicitly specified 1-2 sota methods to compare, rather than giving a list of models, which is not clear and makes it impossible for us to complete.
> > > > > > >
> > > > > > > ● **Parameter search concerning**
> > > > > > >
> > > > > > > 1）Regarding the emphasized original review *'Results can vary significantly for the same parameters on different platforms. Therefore, a fair approach would involve setting a consistent search space for all methods on the same platform and determining the best parameters for each model on various tasks using a validation set.'*, **all of our model parameter selections were based on the same platform and determining the best parameters for each model on various tasks based on the descent of the loss function in the training set and the prediction error in the validation set.**
> > > > > > >
> > > > > > > 2）As for the issue of *hyperparameter coverage*, our model performs much better than other models in Table 2, which is difficult to surpass by adjusting parameters and also demonstrates the superiority of the model. As shown in Table 2, Compared with the optimal baseline, CRAFT improves by at least {0.0447, 0.0166, 0.0063} and {0.0112, 0.0205, 0.0064} in MAE and wMAP E metrics in the prediction window of {7, 14, 30}, **which is difficult to surpass by adjusting parameters.**
> > > > > > >
> > > > > > > 3）In addition, **the parameter space search done in the current paper is the hyperparameter that is considered to be more important after experimental verification.** And, we read the informer paper, which only attempts to search part of the space for Input length, Sampling Factor, and Stacking Combination, and does not search for parameters such as 'e_layer' and 'd_layer'.
> > > > > > >
> > > > > > > Based on this, we look forward to you re-evaluating our work. Thank you.

---

### Official Review · Reviewer_5UPa · 2024-11-03

**Soundness:** 2
**Presentation:** 3
**Contribution:** 2
**Rating:** 5
**Confidence:** 3

**Summary:**

The paper introduces CRAFT (CRoss-Future Behavior Awareness-based Time Series Forecasting), an approach designed to enhance time series forecasting (TSF) in e-commerce by leveraging what the authors term Cross-Future Behavior (CFB).
The authors define CFB and introduce it as a novel feature in TSF. Unlike traditional features that only use historical data to predict future trends, CFB includes partially observable future behavior, which can provide early indicators of upcoming trends.
CRAFT integrates CFB through a structured, multi-module framework, including the Koopman Predictor Module (KPM), Internal Trend Mining Module (ITM), and External Trend Guide Module (ETG).
To further enhance prediction accuracy, CRAFT incorporates a loss function that accounts for upper and lower demand limits.
The authors validate CRAFT's effectiveness with offline experiments and online A/B tests, showing improvements over state-of-the-art baselines in both error metrics (MAE, RMSE, wMAPE) and application-specific metrics like Inventory Waste Rate (IWR) and Proportion of Hotels with Depleted Inventory (PHDI).
Overall, CRAFT aims to leverage future behavior patterns effectively to improve TSF, particularly in scenarios where demand is partially predictable through pre-booked actions or similar forward-looking behaviors.

**Strengths:**

S1.
The definition of CFB expands the traditional understanding of time series features by including elements that are observable in advance but affect future outcomes.

S2.
This work is evident in its robust methodology and empirical validation.
The authors employ a well-structured framework composed of three distinct modules—KPM, ITM, and ETG—each addressing specific challenges associated with CFB and time series forecasting.

S3.
This work's demonstrated improvement in forecasting accuracy can have practical implications for businesses, aiding in better resource allocation and inventory management.

**Weaknesses:**

W1.
First of all, while the introduction of CFB is a strong point, the paper could benefit from a broader exploration of its implications and applications.
The current formulation primarily focused on e-commerce and hotel booking scenarios.

W2.
The experimental section could be improved in several ways.
For example, conducting longitudinal studies to evaluate how CRAFT performs over extended periods or under different market conditions would add depth to the findings.

W3.
- A systematic sensitivity analysis could be conducted to understand how hyperparameter variations affect the model’s predictions. This would help practitioners better tune CRAFT for their specific applications.

- Providing specific guidelines for selecting hyperparameters based on data characteristics could enhance the practical utility of the model.

**Questions:**

Q1.
How do you envision CFB being applied in domains beyond e-commerce?
Could you provide specific examples or potential use cases in different sectors?

Q2.
Have you considered conducting longitudinal studies to evaluate the stability and performance of CRAFT over time?
What are the challenges you foresee in such an analysis?

**Details Of Ethics Concerns:**

Using consumer booking data to derive Cross-Future Behavior (CFB) raises concerns about data privacy and the ethical implications of utilizing sensitive consumer information.
If the data is not anonymized or consumer consent is not clearly obtained, this could lead to privacy violations.

---

> ### Author Response · Authors · 2024-11-28
> **Response to Reviewer 5UPa (Part 1/n)**
>
> > Summary
>
> We thank reviewer for reviewing our manuscript and pointing out a set of significiant suggestions. These suggestions improve the quality of our manuscript significantly. We have responded to the reviewer's comments below and have addressed them accordingly in the uploaded revision of the paper. **Given the improvement suggestion by all reviewers, we have re-modified our manuscript and uploaded the revised version. We look forward to the reviewer's reassessment of our manuscript.**
>
> > **Response to Weakness 1**: broader exploration of its implications and applications
>
> Thanks for your significiant suggestion. Our proposed CFB features and CRAFT method is indeed applicable to forecasting problems with advance operation. **Currently, we only collect suitable experimental dataset in e-commerce area. Therefore, we organize our paper around e-commerce.** In the origial manuscript, we have pointed this limitation in the conclusion sections as: "As current public available dataset (e.g., ETT, ECL, Weather) does not contain the CFB feature or similar feature, we only verify CRAFT on our dataset. In the future, we will further collect related data to form series benchmark dataset. In addition, we will explore the application of CRAFT on more scenarios."
>
> In the revised pdf, we have discussed the application of CFB features and CRAFT method in other area and supplemented the content to conclusion section: "This paper only explores the application of CFB and CRAFT on e-commerce area. Electricity demand, stock price, and disease spread forecasting typically do not have clear lead-up operation events, making it difficult to apply our method. In addition, current public available dataset (e.g., ETT, ECL, Weather) does not contain the CFB feature or similar feature, we only verify CRAFT on our dataset limited by this actual situation. n the future, we will further collect related data to form series benchmark dataset. Moreover, we will continue to generalize the definition of CFB to enable CRAFT method can be applied more broadly."
>
> > **Response to Weakness 2**: conduct longitudinal studies and studies under different market conditions
>
> Thanks for your suggestion. Actually, we have already conducted longitudinal studies to evaluate how CRAFT performs over extended periods and under different market conditions:
>
> **Longitudinal studies**: Section 5.2.1 (Comparison with Baselines) is the longitudinal studies. We conduct experiment  with prediction window length of P 7, 14, 30 respectively, correspond one-to-one with the look back window L 30, 90, 180.  As for why the window is not long enough, such as a full year, it is because predictions made 7-30 days in advance are sufficient for sellers to adjust prices and inventory. Predictions made too far in advance do not have significant meaning for actual business operations.
>
> **Under different market conditions**: Section 5.2.3 (Cases Studies) verifies that our proposed method is effective in capturing future trends, under different trends for validation. Section 5.3 (Online A/B Test) is the experiment under different market conditions, including 2023 Mid-autumn, 2023 National Day, 2024 New Year's Day, and 2024 Spring Festival. We are sorry that our introduction in the first draft of the paper is not clear enough.
>
> > **Response to Weakness 3**: sensitivity analysis and the how hyperparameters are selected?
>
> Thanks for your sincere suggestion. The response regarding sensitivity analysis and hyperparameter selection is as follows:
>
> **Sensitivity analysis**:
> Actually, we have already presented the sensitivity analysis studies in Section E.2 Hyper Parameter Analysis. We offered the sensitivity analysis of the loss coefficient and the parameter in solving koopman matrix that has a significant impact on CRAFT. In addition, we will provide the search space and optimal values for other model parameters in Appendix F Experimental Configuration.
>
> **Guidelines for selecting hyperparameters**: We use the loss function of train set and the prediction accuracy of validation set to guide the selection of hyperparameters, which is a common method in machine learning. Firstly, the time-series data is divided into training set, validation set, and testing set along the time dimension, which can avoid the problem of data traversal. The model is trained on the training set and hyperparameters are selected based on the descent of the loss function in the training set and the prediction error in the validation set. Cheng et al. [1] used the similar method. We will also supplement the guidelines on hyperparameter selection in Appendix F Experimental Configuration.
>
> [1] Cheng H, Wen Q, Liu Y, et al. RobustTSF: Towards Theory and Design of Robust Time Series Forecasting with Anomalies[C]//The Twelfth International Conference on Learning Representations.2024.

---

> > ### Author Response · Authors · 2024-11-28
> > **Response to Reviewer 5UPa (Part 2/n)**
> >
> > > **Response to Question 1**: the use of CFB in other areas
> >
> > Thanks for your significiant suggestion. Our proposed CFB features and CRAFT method is indeed applicable to forecasting problems with advance operation. **Currently, we only collect suitable experimental dataset in e-commerce area. Therefore, we organize our paper around e-commerce.** In the origial manuscript, we have pointed this limitation in the conclusion sections as: "As current public available dataset (e.g., ETT, ECL, Weather) does not contain the CFB feature or similar feature, we only verify CRAFT on our dataset. In the future, we will further collect related data to form series benchmark dataset. In addition, we will explore the application of CRAFT on more scenarios."
> >
> > In the revised pdf, we have discussed the application of CFB features and CRAFT method in other area and supplemented the content to conclusion section: "This paper only explores the application of CFB and CRAFT on e-commerce area. Electricity demand, stock price, and disease spread forecasting typically do not have clear lead-up operation events, making it difficult to apply our method. In addition, current public available dataset (e.g., ETT, ECL, Weather) does not contain the CFB feature or similar feature, we only verify CRAFT on our dataset limited by this actual situation. n the future, we will further collect related data to form series benchmark dataset. Moreover, we will continue to generalize the definition of CFB to enable CRAFT method can be applied more broadly."
> >
> > > **Response to Question 2**: conduct longitudinal studies and studies under different market conditions
> >
> > Thanks for your suggestion. Actually, we have already conducted longitudinal studies to evaluate how CRAFT performs over extended periods and under different market conditions:
> >
> > **Longitudinal studies**: Section 5.1.2 (Comparison with Baselines) is the longitudinal studies. We conduct experiment  with prediction window length of $P \in \{7, 14, 30\}$ respectively, correspond one-to-one with the look back window $L \in \{30, 90, 180\}$.  As for why the window is not long enough, such as a full year, it is because predictions made 7-30 days in advance are sufficient for sellers to adjust prices and inventory. Predictions made too far in advance do not have significant meaning for actual business operations.
> >
> > **Under different market conditions**: Section 5.2.3 (Cases Studies) verifies that our proposed method is effective in capturing future trends, under different trends for validation. Section 5.3 (Online A/B Test) is the experiment under different market conditions, including 2023 Mid-autumn, 2023 National Day, 2024 New Year's Day, and 2024 Spring Festival. We are sorry that our introduction in the first draft of the paper is not clear enough.
> >
> > ---
> >
> > > **Response to Ethics Concerns**:
> >
> > Thank you for raising this crucial point regarding the privacy and ethical considerations of using consumer data. We are fully aware of the importance of data privacy and the ethical implications associated with handling sensitive consumer information.
> >
> > In response to your concern, we would like to make absolutely sure that the data is anonymized and consumer consent is obtained when the data used in this paper is released. We have taken the following steps to address the issues you've highlighted:
> >
> > 1. **Data Anonymization** : We have implemented stringent measures to ensure that all data used in our paper is anonymized. This process involves removing any personally identifiable information to protect the privacy of our consumers.
> > 2. **Consumer Consent** : Our experimental data consists of hotel booking volumes, aggregated by different hotel dimensions. Therefore, our experimental data does not involve any specific consumer privacy issues.
> > 3. **Seller Consent** : We have obtained explicit consent from all sellers whose data is included in our paper. This consent is informed, meaning that participants are fully aware of how their data will be used and for what purposes.
> > 4. **Ethical Review** : Our dataset has undergone a thorough ethical review by an independent committee within our organization. This review process is designed to identify and mitigate any potential ethical issues, ensuring that our research adheres to the highest standards of ethical conduct.
> > 5. **Ongoing Compliance** : We are committed to ongoing compliance with data protection regulations and ethical guidelines. This includes regular audits and updates to our data handling procedures to ensure they remain aligned with current best practices and legal requirements.
> >
> > We believe that these measures not only address the concerns you have raised but also demonstrate our dedication to conducting research that is both scientifically rigorous and ethically sound.

---

> > > ### Author Response · Authors · 2024-12-02
> > > **Response to Reviewer 5UPa (Part 3/n)**
> > >
> > > Dear Reviewer 5UPa,
> > >
> > > Thank you again for your time and effort in reviewing our paper.
> > >
> > > As the new Discussion Period ends on December 2nd at 24:00 AoE, we kindly remind you to review our revised manuscript and responses. We are eagerly awaiting your feedback.

---

> > > > ### Author Response · Authors · 2024-12-03
> > > > **Response to Reviewer 5UPa (Part 4/n)**
> > > >
> > > > Dear Reviewer 5UPa,
> > > >
> > > > Due to the discussion between the author and the reviewer being near its conclusion, we would like to know whether our response has addressed your main concerns. If so, we kindly ask for your reconsideration of the score. Should you have any further advice on the paper and/or our rebuttal, please let us know and we will be more than happy to engage in more discussion and paper improvements.
> > > >
> > > > Thank you so much for devoting time to improving our work!

---

> > > > > ### Comment · Reviewer_5UPa · 2024-12-03
> > > > > **Response to Authors' Rebuttals by Reviewer 5UPa**
> > > > >
> > > > > Thanks to the authors for their rebuttals.
> > > > > My concerns are partially addressed.
> > > > > However, as the authors mentioned, this work's applicability is limited to the e-commerce domain, which makes its implications less impressive and significant.
> > > > > I will keep my scores.

---

> > > > > > ### Author Response · Authors · 2024-12-03
> > > > > > **Response to Reviewer 5UPa (Part 5/n)**
> > > > > >
> > > > > > Dear 5UPa,
> > > > > >
> > > > > > Thanks for your timely responses.
> > > > > > Despite differences in our understanding of the limitations of a paper, we still hope to have a voice in our manuscript.
> > > > > >
> > > > > > We first need to reach a consensus that **every work has its own limitations**. Even the outstanding Transformer, in its initial version, was more suitable for sequential data such as speech and text.
> > > > > >
> > > > > > In ICLR, there are also some works that focus on specific application area, such as [1] and [2].
> > > > > >
> > > > > > Our proposed method is indeed applicable to forecasting problems with advanced operation, not just e-commerce area. However, **we only collect suitable experimental dataset in e-commerce area. Therefore, to improve the fluency of writing, we organize our paper around e-commerce.**
> > > > > >
> > > > > > Finaly, we would like to emphasize our paper's contribution again:
> > > > > > 1) **A High-Quality Dataset**: In this paper, we contribute a high-quality time series forecasting dataset. Once the paper is published, we will release our dataset. This dataset will not only serve as a benchmark for time series forecasting problems but also provide a richer set of features compared to existing datasets.
> > > > > > 2) **CFB features and CRAFT Method**: We define the CFB feature innovatively and apply the CFB feature to time series forecasting for the first time. CFB is a feature discovered from our extensive real case studies and has superior characteristics: the trend of CFB can reflect the prediction target and even the abnormal trend of the target. In addition, we propose a novel framework, namely CRAFT, to realize CFB-based time series forecasting. CRAFT can utilize the trend of CFB to mine the trend of prediction targets.
> > > > > > 3) **Extensive Experiments**: We conduct extensive experiment, including the comparative experiment (Section 5.2.1), the ablation study (Section 5.2.2), the case study (Section 5.2.3), the online A/B test (Section 5.3), visualization of key modules (Appendix E.1), hyper parameter analysis (Appendix E.2), time complexity analysis (Appendix E.3), and using CFB as label (Appendix E.4). These comprehensive and extensive experiments have all validated the effectiveness of our CFB features and CRAFT method.
> > > > > >
> > > > > > Based on this, we look forward to you re-evaluating our work. Thank you.
> > > > > >
> > > > > > [1] Bose A J, Akhound-Sadegh T, Huguet G, et al. Se (3)-stochastic flow matching for protein backbone generation[C]//International Conference on Learning Representations (ICLR), 2024.
> > > > > >
> > > > > > [2] Liu C, Zhou X, Zhu Z, et al. VBH-GNN: Variational Bayesian Heterogeneous Graph Neural Networks for Cross-subject Emotion Recognition[C]//The Twelfth International Conference on Learning Representations (ICLR), 2024.

---

### Official Review · Reviewer_WMRA · 2024-11-05

**Soundness:** 3
**Presentation:** 3
**Contribution:** 3
**Rating:** 6
**Confidence:** 4

**Summary:**

The paper introduces CRAFT (Cross-Future Behavior Awareness based Time Series Forecasting), a novel time series forecasting method designed to enhance technical support for consumer behavior analysis and sales trend prediction in the e-commerce sector. CRAFT focuses on "Cross-Future Behavior" (CFB), which refers to features that occur before the current time but take effect in the future, reflecting future sales trends and increasing the certainty of forecasting.

**Strengths:**

CRAFT is an innovative time series forecasting method that enhances predictive model performance by defining and leveraging Cross-Future Behavior (CFB). It comprises three main modules to address the sparsity and partiality of CFB, as well as to acquire representative trends from higher levels, and calibrates the distribution deviation of forecast results with a demand-constrained loss. CRAFT has demonstrated exceptional performance on real-world datasets, significantly improving prediction accuracy over existing techniques, and has been successfully applied to practical scenarios such as online hotel inventory negotiations. Moreover, its design facilitates further research and development, exploring its potential application in a variety of scenarios.

**Weaknesses:**

The research on CRAFT, while presenting a significant advancement in time series forecasting, does have certain limitations that can be discussed in terms of real-world data complexity, redundancy, interpretability, and data transferability:

1. **Real-World Data Complexity and Variability:**
   - Real-world data is often characterized by noise, outliers, and non-stationarity, which can affect the model's ability to learn accurate patterns. CRAFT may struggle to capture these complex dynamics, especially if they are not well-represented in the training data.

2. **Data Redundancy:**
   - The incorporation of CFB along with other features might lead to data redundancy, which could impact the model's efficiency and potentially its accuracy. There is a need for feature selection techniques to ensure that the information used is diverse and non-redundant, focusing on the most predictive signals.

3. **Interpretability:**
   - The black-box nature of some components in CRAFT, such as the Koopman Predictor Module, can make it difficult to interpret how predictions are made, which is a critical aspect for stakeholders who need to understand the reasoning behind the model's outputs.

4. **Data Transferability:**
   - The model's performance may not be consistent across different domains or datasets due to the unique characteristics of each data environment. The reliance on CFB, which may not be universally applicable, could limit the model's transferability to other contexts where such behavior patterns do not exist or are less pronounced.

5. **Generalizability:**
   - The study primarily focuses on e-commerce data, and it is unclear how well CRAFT would perform in other industries or with different types of time series data. Further testing and validation on a diverse range of datasets are needed to establish the model's generalizability.

6. **Scalability:**
   - The paper does not extensively address the scalability of the CRAFT model, particularly in handling large-scale datasets that are common in many real-world applications. The computational complexity and resource requirements could be a limiting factor.

7. **Robustness to Changing Conditions:**
   - The model's robustness to changing conditions, such as shifts in consumer behavior or market dynamics, is not fully explored. Real-world applications require models that can adapt to new trends and patterns over time.

8. **Dependency on High-Quality Data:**
   - CRAFT's performance is likely to be highly dependent on the quality and granularity of the input data. In scenarios where data is limited or of poor quality, the model's effectiveness may be compromised.

Addressing these limitations would involve further research into robust data preprocessing techniques, enhancing model interpretability, and conducting cross-domain validations to ensure that CRAFT can meet the challenges of real-world data analytics.

**Questions:**

When evaluating the experimental data and model practices of CRAFT papers, the following are several key ethical and practical issues:
1. Ethical and Privacy Issues of Data:
-Experimental data must ensure compliance with ethical standards, especially regarding the protection of user privacy. Any data containing personally identifiable information should be anonymized to prevent privacy breaches.
2. Data ownership issue :
-The data used needs to have clear ownership and usage rights. Researchers must ensure that they have the right to use this data and that the use of the data complies with the regulations and laws of the data source.
3.   Use publicly available benchmark data for evaluation  :
-To improve the transparency and comparability of model evaluation, publicly available benchmark datasets can be used for evaluation. This helps to validate the generalization ability and performance of the CRAFT model on different datasets.
4.   Generalization ability of the model  :
-The generalization ability of a model is a key factor in evaluating its practicality. The CRAFT model needs to be tested on multiple different datasets to ensure that it not only performs well on specific datasets, but also can be widely applied in various scenarios.
5.   Ensure the stability and security of the model  :
-The stability and security of the model are crucial for its practical application. Strict testing and validation are required to ensure that the model maintains performance under various conditions and is resistant to potential security threats.

**Details Of Ethics Concerns:**

Does the experimental data in the paper have ethical and moral issues, is it private data, is the data authentic, and can it be evaluated using publicly available benchmark data? What is the generalization ability of the model? How to ensure the stability and security of the model?

---

> ### Author Response · Authors · 2024-11-28
> **Response to Reviewer WMRA (Part 1/n)**
>
> > Summary
>
> We thank reviewer for reviewing our manuscript and pointing out a set of significiant questions. We have responded to the reviewer's comments below and have addressed them accordingly in the uploaded revision of the paper. **Given the improvement suggestion by all reviewers, we have re-modified our manuscript and uploaded the revised version. We look forward to the reviewer's reassessment of our manuscript.**
>
> > **Response to Weaknesses**:
>
> Thank you for your insightful and detailed comments. We appreciate the opportunity to address the limitations of our work. Here are the responses to each point:
>
> 1. **Real-World Data Complexity and Variability**:
>    We acknowledge that real-world data often contains noise, outliers, and non-stationarity, which can indeed pose challenges for any forecasting model. To verify CRAFT's ability to handle such complexities, we conduct comprehensive experiments on different data.  In the comparative experiment (Section 5.2.1), we conduct experiment on data with different window length. In the case study (Section 5.2.3), we conduct experiment on different data with different trends for validation. In the online A/B test (Section 5.3), we conduct experiment during different holiday, including 2023 Mid-autumn, 2023 National Day, 2024 New Year's Day, and 2024 Spring Festival. These diverse experiments on various data sets, along with the rich experimental results, have all validated the superiority of our approach and the diversity of our dataset.
> 2. **Data Redundancy**:
>    The introduction of CFB features indeed increases the dimensionality of the features, which is also the main reason why the direct use of CFB features in other existing methods shown in Table 2 is not effective enough. Therefore, to address the issue of fully utilizing CFB features, we propose the CRAFT method. The core idea of CRAFT is to utilize the trend of CFB to mine the trend of time series data to be predicted.
> 3. **Interpretability**:
>    We recognize the importance of interpretability, especially for stakeholders who need to understand the decision-making process. CRAFT is composed of three main parts:  KPM module, ITM module, and ETG module. Each module has a clear function and well-defined input and output results. KPM can extract the key trends of the label and CFB, predicting the label in the prediction window. ITM supplements the unknown area of CFB, making the final prediction of the label in the prediction window. ETG, with a hierarchical structure, can acquire more representative trends from higher levels. Figure 12 visualizes the experiemtal results for every modules. According to Figure 12, we can not only observe the experiemental results after every modules, but also know the consistency of our model's prediction results with the actual labels.
> 4. **Data Transferability**:
>    Our proposed CFB features and CRAFT method is indeed applicable to forecasting problems with advance operation. Currently, we only collect suitable experimental dataset in e-commerce area. We will discuss the application of CFB features and CRAFT method in other area and supplement the content to conclusion section.
> 5. **Generalizability**:
>    Currently, we only collect suitable experimental dataset in e-commerce area. Therefore, we organize our paper around e-commerce. If we can acquire more suitable dataset in other area, we will conduct experiment on other dataset.
> 6. **Scalability**:
>    Due to the using area limitation, we do not conduct experiment on other large-scale datasets. In the revised version of our manuscript, we will discuss the application of CFB features and CRAFT method in other area and supplement the content to conclusion section.
> 7. **Robustness to Changing Conditions**:
>    Actually, our paper has discussed some content related to robustness to changing condition. In the comparative experiment (Section 5.2.1), we conduct experiment on data with different window length. In the case study (Section 5.2.3), we conduct experiment on different data with different trends for validation. In the online A/B test (Section 5.3), we conduct experiment during different holiday, including 2023 Mid-autumn, 2023 National Day, 2024 New Year's Day, and 2024 Spring Festival.
> 8. **Dependency on High-Quality Data**:
>    We understand that the quality of input data is crucial for the effectiveness of CRAFT. We are sorry that we only conduct experiment on our dataset limited by the data condition.

---

> > ### Author Response · Authors · 2024-11-28
> > **Response to Reviewer WMRA (Part 2/n)**
> >
> > > **Response to Question 1 and Question 2**: ethical and privacy issues of data and data ownership issue
> >
> > As we have declared in the abstract parts: **"Our dataset and code will be released after formal publication."**
> >
> > We would like to make absolutely sure that the data is anonymized and consumer consent is obtained when the data used in this paper is released. We have taken the five steps to address the ethical, moral and private issues. 1) *Data Anonymization* : We have implemented stringent measures to ensure that all data used in our paper is anonymized. This process involves removing any personally identifiable information to protect the privacy of our consumers. 2) *Consumer Consent* : Our experimental data consists of hotel booking volumes, aggregated by different hotel dimensions. Therefore, our experimental data does not involve any specific consumer privacy issues. 3) *Seller Consent* : We have obtained explicit consent from all sellers whose data is included in our paper. This consent is informed, meaning that participants are fully aware of how their data will be used and for what purposes. 4) *Ethical Review* : Our dataset has undergone a thorough ethical review by an independent committee within our organization. This review process is designed to identify and mitigate any potential ethical issues, ensuring that our research adheres to the highest standards of ethical conduct. 5) *Ongoing Compliance* : We are committed to ongoing compliance with data protection regulations and ethical guidelines. This includes regular audits and updates to our data handling procedures to ensure they remain aligned with current best practices and legal requirements.
> >
> > > **Response to Question 3**: use publicly available benchmark data for evaluation
> >
> > Our proposed CFB features and CRAFT method is indeed applicable to forecasting problems with advance operation. Currently, we only collect suitable experimental dataset in e-commerce area. Therefore, we organize our paper around e-commerce. We will discuss the application of CFB features and CRAFT method in other area and supplement the content to conclusion section.
> >
> > > **Response to Question 4 and Question 5**: generalization ability of the model, ensure the stability and security of the model
> >
> > Our model have excellent generalization and stability ability. The comprehensive experimental results can verify our model's generalization and stability ability.  In the comparative experiment (Section 5.2.1), we conduct experiment with different window length. In the case study (Section 5.2.3), we conduct experiment on different data with different trends for validation. In the online A/B test (Section 5.3), we conduct experiment during different holiday, including 2023 Mid-autumn, 2023 National Day, 2024 New Year's Day, and 2024 Spring Festival. In addition, we also conduct experiment related to visualization of key modules (Appendix E.1), hyper parameter analysis (Appendix E.2), time complexity analysis (Appendix E.3), and using CFB as label (Appendix E.4). We also conduct experiment on our data with other baseline methods (Table 2). In terms of model security , our model does not use a distributed architecture and does not involve sharing our non-anonymized data and model parameters with third parties.

---

> > > ### Author Response · Authors · 2024-11-28
> > > **Response to Reviewer WMRA (Part 3/n)**
> > >
> > > > **Response to Ethics Concerns**:
> > >
> > > Thank you for raising the ethics problem about our dataset and scientific issues about our model. We will response your series inquiries one by one:
> > >
> > > 1. **Ethical, Moral and Private Issues of Data**:  We would like to make absolutely sure that the data is anonymized and consumer consent is obtained when the data used in this paper is released. We have taken the five steps to address the ethical, moral and private issues. 1) *Data Anonymization* : We have implemented stringent measures to ensure that all data used in our paper is anonymized. This process involves removing any personally identifiable information to protect the privacy of our consumers. 2) *Consumer Consent* : Our experimental data consists of hotel booking volumes, aggregated by different hotel dimensions. Therefore, our experimental data does not involve any specific consumer privacy issues. 3) *Seller Consent* : We have obtained explicit consent from all sellers whose data is included in our paper. This consent is informed, meaning that participants are fully aware of how their data will be used and for what purposes. 4) *Ethical Review* : Our dataset has undergone a thorough ethical review by an independent committee within our organization. This review process is designed to identify and mitigate any potential ethical issues, ensuring that our research adheres to the highest standards of ethical conduct. 5) *Ongoing Compliance* : We are committed to ongoing compliance with data protection regulations and ethical guidelines. This includes regular audits and updates to our data handling procedures to ensure they remain aligned with current best practices and legal requirements.
> > > 2. **Authentic Issues of Data**: The authenticity of the data is fundation of our paper. To address your concern, we would like to provide the following information. 1) *Data Collection Methodology* : In current phase, we do not provide the detail about our data collection process to obey the double-blind reviewing request. Once our paper is accepted, we will provide the detailed data collection methodology. 2) *Endorsement from the Organization*: Our data is collected from a formal organization. The authority and review report of the organization can also endorse the authenticity of our data. 3) *Comprehensive Experimental Results* : In our paper, we make comprehensive experiment, including the comparative experiment (Section 5.2.1), the ablation study (Section 5.2.2), the case study (Section 5.2.3), the online A/B test (Section 5.3), visualization of key modules (Appendix E.1), hyper parameter analysis (Appendix E.2), time complexity analysis (Appendix E.3), and using CFB as label (Appendix E.4). We also conduct experiment on our data with other baseline methods (Table 2). These comprehensive experiments also very the autheticity of our data.
> > > 3. **Evaluate Using Publicly Available Benchmark Data** : Our proposed CFB features and CRAFT method is indeed applicable to forecasting problems with advanced operation. In some authoritative dataset, such as M5 dataset [1], there are also auxiliary information like promotions and vacations. However, the auxiliary data in M5 dataset has significant difference with the CFB features. Therefore, we can not conduct experiment on M5 dataset. Currently, we have not found other suitable publicly available benchmark dataset.
> > > 4. **The Generalization, Stability, and Security of the model**: Our model have excellent generalization and stability ability. The comprehensive experimental results can verify our model's generalization and stability ability.  In the comparative experiment (Section 5.2.1), we conduct experiment with different window length. In the case study (Section 5.2.3), we conduct experiment on different data with different trends for validation. In the online A/B test (Section 5.3), we conduct experiment during different holiday, including 2023 Mid-autumn, 2023 National Day, 2024 New Year's Day, and 2024 Spring Festival. In addition, we also conduct experiment related to visualization of key modules (Appendix E.1), hyper parameter analysis (Appendix E.2), time complexity analysis (Appendix E.3), and using CFB as label (Appendix E.4). We also conduct experiment on our data with other baseline methods (Table 2). In terms of model security , our model does not use a distributed architecture and does not involve sharing our non-anonymized data and model parameters with third parties.
> > >
> > > [1] Chen S A, Li C L, Yoder N, et al. Tsmixer: An all-mlp architecture for time series forecasting[J]. arXiv preprint arXiv:2303.06053, 2023.

---

### Author Response · Authors · 2024-11-28
**Revision Summary**

Dear reviewers, AC, and PC,

We sincerely thank you for your time and suggestions. We are grateful that the postive feedback to the innovation, methodology, and experimental results of our paper. We revised the paper according to the reviewers' suggestions. The revised version of the paper is uploaded  with the modification using blue font. Thank you again for your time and significant suggestions. We sincerely our response can address all you concerns. If you have any questions, please let us know.

We would like to emphasize our paper's contribution again:

- **A High-Quality Dataset**: In this paper, we contribute a high-quality time series forecasting dataset. Once the paper is published, we will release our dataset. This dataset will not only serve as a benchmark for time series forecasting problems but also provide a richer set of features compared to existing datasets.
- **CFB features and CRAFT Method**: We define the CFB feature innovatively and apply the CFB feature to time series forecasting for the first time. CFB is a feature discovered from our extensive real case studies and has superior characteristics: the trend of CFB can reflect the prediction target and even the abnormal trend of the target. In addition, we propose a novel framework, namely CRAFT, to realize CFB-based time series forecasting. CRAFT can utilize the trend of CFB to mine the trend of prediction targets.
- **Extensive Experiments**: We conduct extensive experiment, including the comparative experiment (Section 5.2.1), the ablation study (Section 5.2.2), the case study (Section 5.2.3), the online A/B test (Section 5.3), visualization of key modules (Appendix E.1), hyper parameter analysis (Appendix E.2), time complexity analysis (Appendix E.3), and using CFB as label (Appendix E.4). These comprehensive and extensive experiments have all validated the effectiveness of our CFB features and CRAFT method.

---

### Meta-Review · Area_Chair_Lngx · 2024-12-20

**Metareview:**

This paper proposes a method that leverages Cross-Future Behavior features for time series forecasting in e-commerce.
It introduces an interesting concept and provides comprehensive empirical evaluations within its scope, but concerns about its generalizability, originality, and experimental rigor outweigh its contributions.
Addressing the lack of SOTA comparisons, issues with parameter exploration, the limited scope of applicability, and dataset constraints is encouraged in future.

**Additional Comments On Reviewer Discussion:**

While the discussion during the rebuttal period clarified some points, fundamental concerns regarding generalizability, innovation, and experimental rigor remain unresolved.

---

### Decision · Program_Chairs · 2025-01-22

Reject